

# Meteorology-driven variability of air pollution (PM1) revealed with explainable machine learning

Roland Stirnberg[1,2], Jan Cermak[1,2], Simone Kotthaus[3], Martial Haeffelin[3], Hendrik Andersen[1,2], Julia Fuchs[1,2], Miae Kim[1,2], Jean-Eudes Petit[4], and Olivier Favez[5]

[1]Institute of Meteorology and Climate Research, Karlsruhe Institute of Technology (KIT), Karlsruhe, Germany
[2]Institute of Photogrammetry and Remote Sensing, Karlsruhe Institute of Technology (KIT), Karlsruhe, Germany
[3]Institut Pierre Simon Laplace, École Polytechnique, CNRS, Institut Polytechnique de Paris, Palaiseau, France
[4]Laboratoire des Sciences du Climat et de l'Environnement, CEA/Orme des Merisiers, Gif sur Yvette, France
[5]Institut National de l'Environnement Industriel et des Risques, Parc Technologique ALATA, Verneuil en Halatte, France

**Correspondence:** Roland Stirnberg (Roland.Stirnberg@kit.edu)

**Abstract.** Air pollution, in particular high concentrations of particulate matter smaller than 1 $\mu$m in diameter (PM1), continues to be a major health problem, and meteorology is known to substantially contribute to atmospheric PM concentrations. However, the scientific understanding of the complex mechanisms leading to high pollution episodes is inconclusive, as the effects of meteorological variables are not easy to separate and quantify. In this study, a novel, data-driven approach based on empirical relationships is used to characterise the role of meteorology on atmospheric concentrations of PM1. A tree-based machine learning model is set up to reproduce concentrations of speciated PM1 at a suburban site southwest of Paris, France, using meteorological variables as input features. The contributions of each meteorological feature to modeled PM1 concentrations are quantified using SHapley Additive exPlanation (SHAP) regression values. Meteorological contributions to PM1 concentrations are analysed in selected high-resolution case studies, contrasting season-specific processes. Model results suggest that winter pollution episodes are often driven by a combination of shallow mixed layer heights (MLH), low temperatures, low wind speeds or inflow from northeastern wind directions. Contributions of MLHs to the winter pollution episodes are quantified to be on average ∼5 $\mu$g/m$^3$ for MLHs below <500 m agl. Temperatures below freezing initiate formation processes and increase local emissions related to residential heating, amounting to a contribution of as much as ∼9 $\mu$g/m$^3$. Northeasterly winds are found to contribute ∼5 $\mu$g/m$^3$ to total PM1 concentrations (combined effects of u- and v-wind components), by advecting particles from source regions, e.g. central Europe or the Paris region. However, in calm conditions (i.e. wind speeds < ∼2 m/s), the lack of dispersion leads to increased PM1 concentrations by ∼3 $\mu$g/m$^3$. Unusually high PM1 concentrations in summer are generally lower compared to winter peak concentrations, and are characterised by a higher content of organics. Meteorological drivers of summer peak PM1 concentrations are temperatures above ∼25 °C (contributions of up to ∼2.5 $\mu$g/m$^3$), dry spells of several days (maximum contributions of ∼1.5 $\mu$g/m$^3$) and wind speeds below ∼2 m/s (maximum contributions of ∼3 $\mu$g/m$^3$ ). High-resolution case studies show a large variability of processes, which together lead to high PM1 concentrations. Processes vary even within seasons. A high pollution episode in January 2016 is shown to be driven by a drop in temperature (maximum contributions of 11 $\mu$g/m$^3$), which enhances formation of secondary inorganic aerosols (SIA) and likely causes an increase in local wood-burning emissions. In contrast, during December 2016, high PM1 concentrations are caused mainly by a shallow





MLH and low wind speeds. It is shown that an observed decrease in pollution levels is linked to a change in wind direc-

tion, advecting cleaner, maritime air to the PM measurement site (combined contributions of u- and v-wind-components of $\sim$-4 $\mu$g/m$^3$). The application of SHAP regression values within a machine learning framework presents a novel and promising way of analysing observational data sets in environmental sciences.

## 1 Introduction

Air pollution has serious implications on human well-being, including deleterious effects on the cardiovascular system and the lungs (Hennig et al., 2018; Lelieveld et al., 2019), and an increased number of asthma seizures (Hughes et al., 2018). This includes particles smaller than 1 $\mu$m in diameter (PM1), which are associated with fits of coughing (Yang et al., 2018) and an increase in emergency hospital visits (Chen et al., 2017). The adverse health effect lead to an increase in mortality of people exposed to high particle concentrations (Samoli et al., 2008, 2013; Lelieveld et al., 2015). People living in urban areas are

particularly affected by poor air quality and with increasing urbanization, their number is projected to grow (Baklanov et al., 2016; Li et al., 2019b). These developments have motivated several countermeasures to improve air quality. Proposed efforts to reduce anthropogenic particle emissions include partial traffic bans (Su et al., 2015; Dey et al., 2018) and the reduction of solid fuel use for domestic heating (Chafe et al., 2014). Although emissions play an important role for PM concentrations in the atmosphere, meteorological conditions related to large-scale circulation patterns as well as local-scale boundary layer pro-

cesses and interactions with the land surface are major drivers of PM variability as well (Cermak and Knutti, 2009; Bressi et al., 2013; Megaritis et al., 2014; Dupont et al., 2016; Petäjä et al., 2016; Yang et al., 2016; Li et al., 2017). It is therefore crucial to take atmospheric and environmental processes into account during the development of efficient pollution mitigation strategies. Wind speed and direction generally have a strong influence on air quality as they determine the advection of pollutants (Petetin et al., 2014; Petit et al., 2015; Srivastava et al., 2018). Limiting the vertical exchange of air masses, the mixed layer height

(MLH) governs the volume of air in which particles typically disperse. Although some authors indicate that mixed layer height cannot be related directly to concentrations of pollutants and that other meteorological parameters and local sources need to be considered (Geiß et al., 2017), a lower MLH can increase PM concentrations as particles are not mixed into higher atmospheric levels and accumulate near the ground (Gupta and Christopher, 2009; Schäfer et al., 2012; Stirnberg et al., 2020).

Higher MLHs in combination with high wind speeds increase atmospheric ventilation processes, thus decreasing near-surface

particle concentrations (Sujatha et al., 2016; Wang et al., 2018). Air temperature can influence PM concentrations in multiple ways, e.g. by modifying the emission of secondary PM precursors such as volatile organic compounds (VOCs) during summer (Fowler et al., 2009; Megaritis et al., 2013; Churkina et al., 2017), and by condensating high saturation vapour pressure compounds such as nitric acid and sulfuric acid (Hueglin et al., 2005; Pay et al., 2012; Bressi et al., 2013; Megaritis et al., 2014). The wet removal of particles by precipitation is known to be an efficient atmospheric aerosol sink (Radke et al., 1980;





Bressi et al., 2013). Although all these atmospheric conditions and processes have been identified as drivers of local air quality, it is usually a complex combination of meteorological and chemical processes that lead to the formation of high-pollution events (Petit et al., 2015; Dupont et al., 2016; Stirnberg et al., 2020). The metropolitan area of Paris is one of the most densely populated and industrialised areas in Europe. Thus, air quality is a recurring issue and has been at the focus of many studies in the past years (Bressi et al., 2014; Petetin et al., 2014; Petit et al., 2015; Dupont et al., 2016; Petit et al., 2017; Srivastava

et al., 2018). Results indicate that the Paris metropolitan region is often affected by mid- to long-range transport of pollutants, as due to the city's flat orography, an efficient horizontal exchange of air masses is frequent (Bressi et al., 2013; Petit et al., 2015). High-pollution events commonly occur in late autumn, winter, and early spring. Often, these episodes are characterised by stagnant atmospheric conditions and a combination of local contributions, e.g. traffic emissions, residential emissions, or regionally transported particles, e.g. ammonium nitrates from manure spreading, or sulfates from point sources (Petetin et al.,

2014; Petit et al., 2014, 2015; Srivastava et al., 2018). High-pressure conditions with air masses originating from continental Europe (Belgium, Netherlands, West Germany) are generally associated with an increase in particle concentrations, especially of secondary inorganic aerosols (SIA, Bressi et al. (2013); Srivastava et al. (2018). The regional contribution has been found to be in the range of 70 % for background concentrations in Paris of particles with a diameter smaller 2.5 $\mu$m (Petetin et al., 2014). Hence the variability between high-pollution episodes in terms of timing, sources and meteorological boundary conditions is

considerable (Petit et al., 2017). Previous approaches to determine meteorological drivers of air pollution included, for example, the use of chemical transport models (CTMs), which, however, require comprehensive knowledge on emission sources and secondary particle formation pathways and are associated with considerable uncertainties (Sciare et al., 2010; Petetin et al., 2014; Kiesewetter et al., 2015). Further methods rely on data exploration, e.g. the statistical analysis of time-series (Dupont et al., 2016), which can be coupled with positive matrix factorization (PMF, Paatero and Tapper, 1994) to derive PM sources

(Petit et al., 2014; Srivastava et al., 2018). To take into account the interconnected nature of PM drivers, multivariate statistical approaches such as principal component analysis (PCA) have been applied (Chen et al., 2014; Leung et al., 2017). In recent years, machine learning techniques have been increasingly used to expand the analysis of PM concentrations with respect to meteorology, allowing to retrace general patterns (Hu et al., 2017; Grange et al., 2018).

Here, the multivariate and highly interconnected nature of the processes determining local PM1 concentrations is captured and

analysed in a data-driven way, capturing empirical relationships between PM1 concentrations and meteorological parameters. A state-of-the-art explainable machine learning model is set up to reproduce the variability of PM1 concentrations, with the objective of advancing the process understanding of the complex mechanisms that lead to high pollution events in Paris. Localised and individualised attributions of feature contributions are performed using SHapley Additive exPlanation regression (SHAP) values (Lundberg and Lee, 2017; Lundberg et al., 2018a, 2020). With the use of SHAP values, a detailed insight to

the decisions of the statistical model can be provided, hence allowing an advancement of previous ML approaches (Friedman, 2001; Lundberg et al., 2018a). The attribution of local (i.e. situation-based) statistical feature contributions enables quantitative estimates of drivers of PM concentrations and allows to infer on processes driving PM concentrations at high temporal resolution. Meteorological effects on speciated PM1 concentrations are quantified and the roles of the most critical atmospheric variables for driving peak particle concentrations are highlighted. The improved scientific understanding of processes is crucial





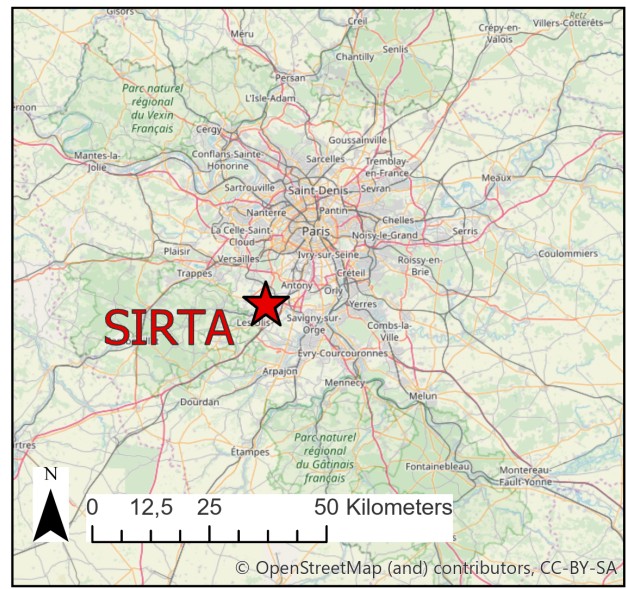

**Figure 1.** Location of the SIRTA supersite southwest of Paris. © OpenStreetMap contributors 2020. Distributed under a Creative Commons BY-SA License.

for the assessment of the effectiveness of measures towards better air quality, and therefore of high value to political decision makers.

## 2   Data sets

Seven years (2012-2018) of meteorological and air quality data from the Site Instrumental de Recherche par Télédétection Atmosphérique (SIRTA, Haeffelin et al., 2005) supersite are the basis of this study. The SIRTA Atmospheric Observatory is located about 25km southwest of Paris (48.713°N and 2.208°E, Fig. 1). This study focuses on day-to-day variations of total and speciated PM1. To separate diurnal effects e.g. the development of the boundary layer during morning hours (Petit et al., 2014; Dupont et al., 2016; Kotthaus and Grimmond, 2018a) from day-to-day variations of PM1, mean concentrations of total and speciated PM1 for the afternoon period 12-15 UTC are considered, when the boundary layer is fully developed. In sections 2.1 and 2.2, the PM1 and meteorological data and preprocessing steps before setting up the machine learning model are described. The applied machine learning model and data analysis techniques are presented in sections 3.1 and 3.2.

### 2.1   Submicron particle measurements

Aerosol chemical speciation monitor (ACSM, Ng et al., 2011) measurements are conducted at SIRTA in the framework of the ACTRIS project. The ACSM provides continuous and near real-time measurements of the major chemical composition of non-refractory submicron aerosols, i.e., organics (Org), ammonium (NH4), sulfate (SO4), nitrate (NO3) and chloride (Cl).





A detailed description of its functionality can be found in Ng et al. (2011). Data processing and validation protocol can be found in Petit et al. (2015) and Zhang et al. (2019). In addition, black carbon (BC) has been monitored by a seven-wavelength Magee Scientific Aethalometer AE31 from 2011 to mid-2013, and a dual-spot AE33 (Drinovec et al., 2015) from mid-2013 onwards. Consistency of both instruments have been checked in Petit et al. (2014). Using the multispectral information, a differentiation into fossil fuel-based BC (BCff) and BC from wood burning (BCwb) is achieved (Sciare et al., 2010; Healy

et al., 2012; Petit et al., 2014; Zhang et al., 2019). Here, the sum of all measured species is assumed to represent the total PM1 content (see Petit et al., 2014, 2015). The consistency of ACSM and Aethalometer measurements is checked by comparing the sum of all monitored species with measurements of a nearby Tapered Element Oscillating Microbalance equipped with a Filter Dynamic Measurement System (TEOM-FDMS). As an additional input to the machine learning model, the average fraction of NO3 of the previous day is added (NO3_frac). Pollution events dominated by NO3 are often linked to regional-scale

events, which depend on anthropogenically-influenced processes in the source regions of NO3 precursors (Petit et al., 2017). This is approximated by the inclusion of the average fraction of NO3 of the previous day, assuming that a high fraction of NO3 indicates the occurrence of such an anthropogenically-influenced regime.

## 2.2 Meteorological data

Meteorological variables included in this study are ambientair temperature (at a height of 2 m, T), relative humidity (RH),

ambient air pressure (AirPres), precipitation, wind speed (ws), wind direction (u, v components) and total incoming solar radiation (TISR). Data are taken from the quality-controlled and 1h averaged re-analysed observation (ReObs) dataset. Further information on the instrumentation used for the acquisition of these variables is provided in Chiriaco et al. (2018). MLH is derived from automatic lidar and ceilometer (ALC) measurements of a Vaisala CL31 ceilometer using the CABAM algorithm Characterising the Atmospheric Boundary layer based on ALC Measurements (Characterising the Atmospheric Boundary

layer based on ALC Measurements, Kotthaus and Grimmond, 2018a, b). Due to an instrument failure, during the period July to mid-November 2016, SIRTA ALC measurements had to be replaced with measurements conducted at the Paris Charles de Gaulle Airport, located northeast of Paris.

Meteorological factors are chosen as input features for the statistical model based on findings of previous studies (see section 1). Meteorological observations are converted to suitable input information for the statistical model (see section 3.1). Wind

speed (ws) is derived from the ReObs u and v components [m/s] and the maximum wind speed of the afternoon period (12-15 UTC) is included in the model. U and v wind components are then normalised to values between 0 and 1, thus only depicting the direction information. To reduce the impact of short-term fluctuation in wind direction, the 3-day mean wind direction is calculated based on the normalised u and v wind components (umean and vmean). Hours since the last precipitation event (Tprec) are counted and used as input to capture the particle accumulation effect between precipitation events (Rost et al.,

2009; Petit et al., 2017).





## 3 Methods

### 3.1 Machine learning model: technique and application

Gradient Boosted Regression Trees (GBRT, used here in a python 3.6.4 environment with the scikit-learn module, Friedman, 2002; Pedregosa et al., 2012) are applied to predict daily total and speciated PM1 concentrations. As a tree-based method,
GBRTs use a tree regressor, which sets up decision trees based on a training data set. The trees split the training data along decision nodes, creating homogeneous subsamples of the data by minimizing the variance of each subsample. For each subsample, regression trees fit the mean response of the model to the observations (Elith et al., 2008). To increase confidence in the model outputs, decision trees are combined to form an ensemble prediction. Trees are sequentially added to the ensemble (Elith et al., 2008; Rybarczyk and Zalakeviciute, 2018) and each new tree is fitted to the predecessor's previous residual error using
gradient descent (Friedman, 2002). This is an advantage of GBRT over standard ensemble tree methods (e.g. Random Forests (RF), Just et al., 2018) as trees are built systematically and fewer iterations are required (Elith et al., 2008). Characteristics of the meteorological training data set with respect to observed total and speciated PM1 concentrations are conveyed to the statistical model. The learned relationships are then used for model interpretation and to produce estimates of PM1 based on unseen meteorological data to test the model. The architecture of the statistical model is determined by the hyperparameters,
e.g. the number of trees, the maximum depth of each tree (i.e., the number of split nodes on each tree) and the learning rate (i.e., the magnitude of the contribution of each tree to the model outcome, which is basically the step size of the gradient descent). The hyperparameters are tuned by executing a grid search, systematically validating testing previously defined hyperparameter combinations and determining the best combination via a three-fold cross validation. Note that PM1 data is not normally distributed, i.e. there is more data available for mid-range concentrations. To avoid that the model primarily optimizes
its predictions on these values, a least-squares loss function was chosen. This loss function is more sensitive to higher PM1 values (i.e. outliers of the PM1 data distribution), as it strongly penalises high absolute differences between predictions and observations. Accordingly, the model is adjusted to reproduce higher concentrations as well.

For each PM species, a specific GBRT model is set up and used for the analysis of meteorological influences on individual PM1 species (see section 4.2). Additionally, a quasi-total PM1 model is used to reproduce the sum of all species at once, which
is used for an analysis of meteorological drivers of high-pollution events (see sections 4.3 and 4.4). Train and test data sets to evaluate each model are created by randomly splitting the full data set. These splits, however, are the same for the species models and the full PM1 model to ensure comparability between the models. Three quarters of the data are used for training and hyperparameter tuning with cross-validation (n=1086), and one quarter for testing (n=363). In addition, the robustness of the model results is tested by repeating this process ten times, resulting in ten models with different train-/ test-splits and
different hyperparameters.

### 3.2 Explaining model decisions: SHapley Additive exPlanation (SHAP) values

While being powerful predictive models, tree-based machine learning methods also have a high interpretability (Lundberg et al., 2020). In order to understand physical mechanisms on the basis of model decisions, the contributions of the meteorological





input features to the model outcome are analysed. Feature contributions are attributed using SHAP values, which allow for an

individualised, unique feature attribution for every prediction (Shapley, 1953; Lundberg and Lee, 2017; Lundberg et al., 2018a, 2020). SHAP values provide a deeper understanding of model decisions than the relatively widely used partial dependence plots (Friedman, 2001; Goldstein et al., 2015; Fuchs et al., 2018; Lundberg et al., 2018a; McGovern et al., 2019; Stirnberg et al., 2020). Partial dependence plots show the global mean effect of an input feature to the model outcome, while SHAP values quantify the feature contribution to each single model output , accounting for multicollinearity. Feature contributions

are calculated from the difference in model outputs with that feature present, versus outputs for a retrained model, without the feature. Since the effect of withholding a feature depends on other features in the model due to interactive effects between the features, differences are computed for all possible feature subset combinations of each data instance (Lundberg and Lee, 2017). Summing up SHAP values for each input feature at a single time step yields the final model prediction. SHAP values can be negative since SHAP values are added to the base value, which is the mean prediction when taking into account all possible

input feature combinations. Negative (positive) SHAP values reduce (raise) the prediction below (above) the base value. The higher the absolute SHAP value of a feature, the more distinct is the influence of that feature on the model predictions. The sum of all SHAP values at one time step yields the final prediction of PM1 concentrations. An example of breaking down a model prediction into feature contributions using SHAP values is shown schematically in Fig. 2. The computation of traditional Shapley Regression values is time consuming, since a large number of all possible feature combinations have to be included.

The SHAP framework for tree-based models allows a faster computation compared to full shapley regression values while maintaining a high accuracy (Lundberg and Lee, 2017; Lundberg et al., 2018a) and is therefore used here. The shap python implementation is used for the computation of SHAP values (https://github.com/slundberg/shap).

Pairwise interactive effects between input features can be estimated using the SHAP approach. Interactive effects are defined as the difference between the SHAP values for one feature when a second feature is present and the SHAP values for the one

feature when the other feature is absent (Lundberg et al., 2018a). SHAP values are a novel tool to better understand multivariate natural systems, in particular when applied in state-of-the-art machine learning models as GBRT. So far, SHAP values have been used in the fields of computer science (see Antwarg et al., 2019) and medical science (Lundberg et al., 2018b; Li et al., 2019a; Lundberg et al., 2020), but have yet to be applied to study environmental systems.

## 4 Results and discussion

### 4.1 Model performance

The performance of the ten model iterations is assessed by comparing the coefficient of determination ($R^2$) and normalised root mean square error (NRSME) for the independent test data that was withheld during the training process (Fig. 3). While the models for BCwb, BCff and total PM1 show small spread, Cl and NO3 exhibit larger variations between model runs (indicated by horizontal and vertical lines in Fig. 3). This suggests that while drivers of variations in BCff concentration are

well covered by the model, this is less so in the case of Cl and NO3. Possible reasons for this are that no explicit information on anthropogenic emissions or chemical formation pathways are included in the models. Still, the model performance indicators





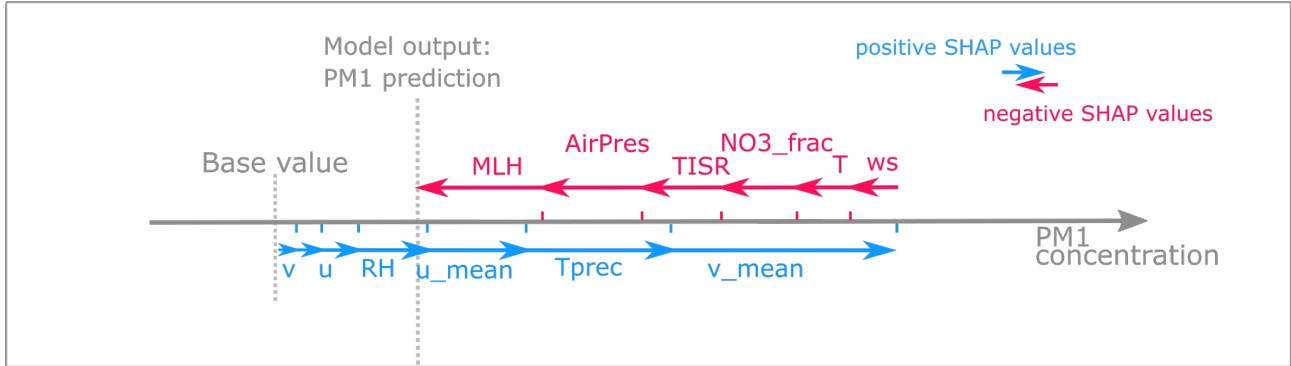

**Figure 2.** Conceptual figure illustrating the interaction of SHAP values and model output. Starting with a base value, which is the mean prediction if all data points are considered, positive SHAP values (blue) increase the final prediction of total and speciated PM1 concentrations, while negative SHAP values (red) decrease the prediction. The sum of all SHAP values for each input feature yields the final prediction. Depending on whether positive or negative SHAP values dominate, the prediction is higher or lower than the base value (Lundberg et al., 2018b). Adapted from https://github.com/slundberg/shap.

highlight that a large fraction of the variations in particle concentrations are explained by the meteorological variables used as model inputs. Performances of model iterations of the species-specific and total PM1 are generally similar, suggesting a robust model outcome.

The mean input feature importance, ordered from high to low, of the total PM1 model run by means of the SHAP feature attribution values is shown in Fig. 4, The NO3 fraction of the previous day has the highest impact on the model, followed by temperature, wind direction information, and MLH. To some extent, NO3 fraction can be related to PM1 mass concentrations (Petit et al., 2015; Beekmann et al., 2015). This means that the higher the PM1 levels one day, the greater the chances of having higher PM1 levels the next day. The impact of the meteorological variables on model decisions is analysed in more detail in

the following.

### 4.2 Contribution of meteorological input features on modelled particle species and total PM1 concentrations

To gain insights on relevant processes governing particle concentrations in the Paris region, the contribution of input features on species and total PM1 concentration outcomes from the statistical model, i.e. the SHAP values, are plotted as a function of absolute feature values (Figs 5-7). The contribution of an input feature to each (local) prediction of the species or total PM1

concentrations is shown while taking into account intra-model variability. Intra-model variability of SHAP values, i.e. different SHAP value attributions for the same feature value within one model, is shown by the vertical distribution of dots for absolute input feature values. Intra-model variability is caused by interactions of the different model input features.

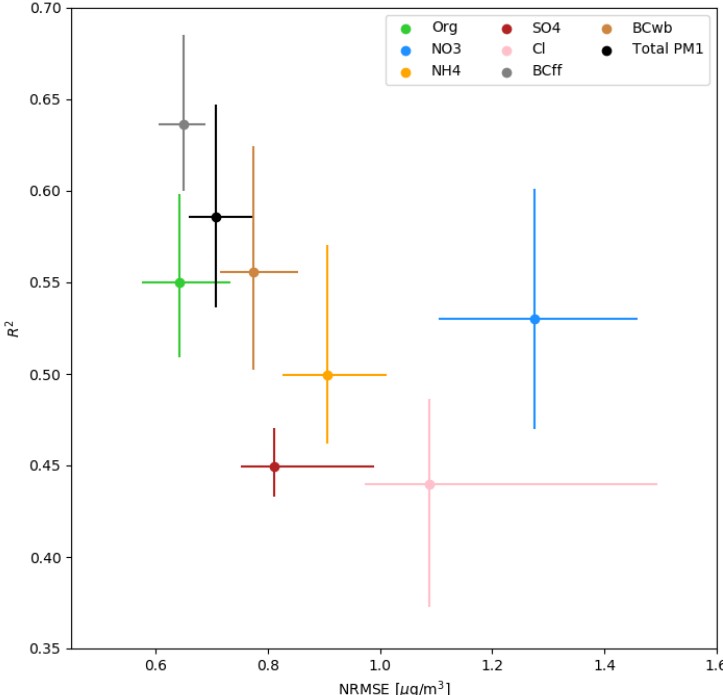

**Figure 3.** Performance indicators for ten model iterations: coefficient of determination $R^2$ against normalised Root Mean Squared Error (NRMSE) for the separate species models and the total PM1 model. Vertical and horizontal lines indicate the maximum spread in $R^2$ and NRMSE, respectively.

### 4.2.1 Contribution of temperature

The impact of ambient air temperature on modelled species concentrations is highly non-linear (Fig. 5). All species show
increased contributions to model outcomes at temperatures below ∼4 °C while the contribution of high temperatures on model outcomes differs substantially between species. The statistical model is able to reproduce well-known characteristics of species concentration variations related to temperature. For example, sulfate formation is enhanced with increasing temperatures (Fig. 5d) due to an increased oxidation rate of SO2 (see Dawson et al., 2007; Li et al., 2017) and strong solar irradiation due to photochemical oxidation (Gen et al., 2019). Dawson et al. (2007) reported an increase of 34 ng/m³K for PM2.5 concentrations
using a CTM. The increase in sulfate at low ambient temperatures is not reported in this study. It is likely linked to increased aqueous phase particle formation in cold and foggy situations (Rengarajan et al., 2011; Petetin et al., 2014; Cheng et al., 2016). Considerable local formation of nitrate at low temperatures (Fig. 5b) is consistent with results from previous studies in western Europe and enhanced formation of ammonium nitrate at lower temperatures (Fig. 5c) by the shifting gas-particle equilibrium

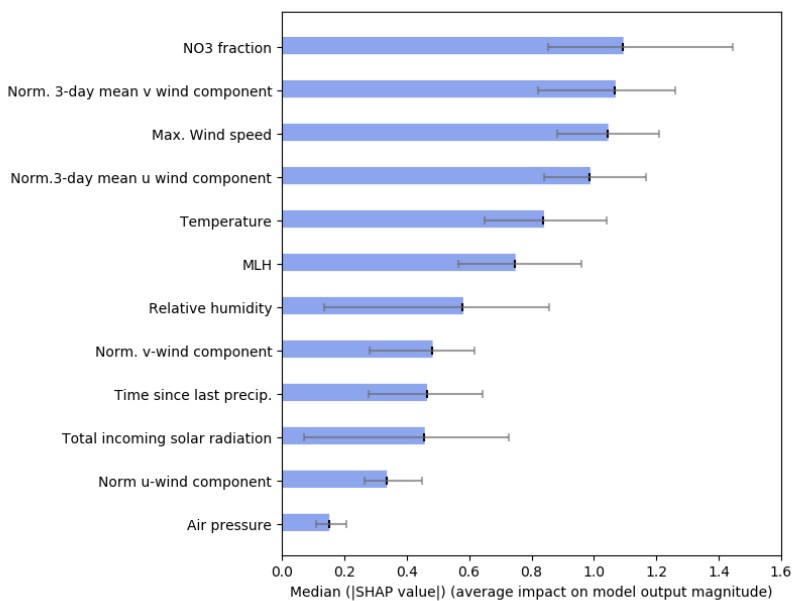

**Figure 4.** Ranked median SHAP values of the model input features, i.e. the average absolute value that a feature adds to the final model outcome, referring to the total PM1 model [$\mu$g/m$^3$] (Lundberg et al., 2018b). Horizontal lines indicate the variability between model runs.

is a well-known pattern (e.g., Clegg et al., 1998; Pay et al., 2012; Bressi et al., 2013; Petetin et al., 2014; Petit et al., 2015). The

increase in organic matter and BCwb concentrations at low temperatures (Fig. 5g) is likely related to the emission intensity, as biomass burning is often used for domestic heating in the study area (Favez et al., 2009; Sciare et al., 2010; Healy et al., 2012; Jiang et al., 2019). In addition, organic matter concentrations are linked to the condensation of semi-volatile organic species at low temperatures (Putaud et al., 2004; Bressi et al., 2013). The sharp increase in modelled concentrations of organics above 25°C (Fig 5a) could be due to enhanced biogenic activity leading to a rise in biogenic emissions and secondary aerosol

formation (Guenther et al., 1993; Churkina et al., 2017; Jiang et al., 2019).

The contribution of temperature on modelled total PM1 concentrations (Fig. 6h) is consistent with the response patterns to changes in temperatures described for the individual species in panels 6a-6g, with positive contributions at both low (<4 °C) and high air temperatures (>25 °C). For temperatures below freezing, the model allocates maximum contributions to modelled total PM1 concentrations of up to 12 $\mu$g/m$^3$. The spread of SHAP values between model iterations is generally higher for low

temperatures (vertical grey bars in Fig. 6), where SHAP values are of greater magnitude, but in all cases the signal contained in the feature contributions far exceeds the spread between model runs.





**Figure 5.** Air temperature SHAP values (contribution of temperature to the prediction of species and total PM1 concentrations [$\mu g/m^3$] for each data instance) vs. absolute air temperature [$^\circ$C]. Inter-model variability of allocated SHAP values is shown as the variance of predicted values between the ten model iterations and plotted as vertical grey bars. The dotted horizontal line indicates the transition from positive to negative SHAP values.





### 4.2.2 Contribution of MLH

Variations in MLH can have a substantial impact on near-surface particle concentrations, as the mixed layer is the atmospheric volume in which the particles are dispersed (see Klingner and Sähn, 2008; Dupont et al., 2016; Wagner and Schäfer, 2017). The

effect of MLH variations on modelled particle concentrations is highly nonlinear for all species (Fig. 6), with the magnitude of the contribution varying by species. Possible reasons for this will be discussed in the following. Similar to the pattern observed for temperature SHAP values, the inter-model variation of predictions is highest for low MLHs since predicted particle concentrations then have the highest variation. Contributions of MLH to predicted particle concentrations are highest for very shallow mixed layers due to the accumulation of particles close to the ground under shallow MLH conditions (Dupont

et al., 2016; Wagner and Schäfer, 2017). A relatively distinct transition from positive contributions during shallow boundary layer conditions (∼0–800 m) towards negative contributions at high MLHs is evident for all species except SO4. Modelled SO4 concentrations show a less distinct response to changes in MLH as they are largely driven by gaseous precursor sources and particle advection, independent of MLH (Pay et al., 2012; Petit et al., 2014, 2015). The accumulation effect under low MLH conditions is hence less important. Furthermore, an increase of SO4 concentrations with higher MLHs (>∼ 1500 m agl) could

be due to a more effective transport of SO4 and its precursor SO2 under high MLH conditions (Pay et al., 2012).

In agreement with results from previous studies focusing on PM10 (Grange et al., 2018; Stirnberg et al., 2020) or PM2.5 (Liu et al., 2018), SHAP values do not change much for MLH above ∼800–900 m, i.e. boundary layer height variations above this level do not influence submicron particle concentrations. Positive contributions for MLHs above ∼800–900 m, as visible in Fig. 6, have been previously reported by Grange et al. (2018), who relate this pattern to enhanced secondary aerosol

formation in a very deep and dry boundary layer. The positive contributions of high MLHs on species that are partly secondarily formed, e.g. SO4 and Org, could be explained explained following this argumentation. However, processes driving the positive contribution of high MLHs on BCff, which is directly emitted to the atmosphere, remain inconclusive. For predicted total PM1 concentrations, the maximum positive contribution of the MLH is as high as 5.5 $\mu$g/m$^3$ while negative contributions can amount to -2 $\mu$g/m$^3$. While the maximum contribution by MLH is lower than the maximum contribution determined for air

temperature, the frequency of shallow MLH is far greater than that of the minimum temperatures that have the largest effect (Figs 5d & 6d). In addition to causing particles to accumulate near the surface, low MLH can also provide effective pathways for local new particle formation. Secondary pollutants, such as ammonium nitrate, are increased at low MLHs as conditions favorable to its formation usually coincide with shallow MLH conditions (i.e., low temperatures, often in combination with high RH, Pay et al., 2012; Petetin et al., 2014; Dupont et al., 2016; Wang et al., 2016). BC concentrations, on the other hand,

are associated with primary emissions, as is a substantial fraction of organic matter (Petit et al., 2015). Hence, the accumulation of these particles during low turbulence conditions can explain the strong influence of MLH on BCwb and BCff.

### 4.2.3 Contribution of wind direction

To analyse the contribution of wind direction to predicted particle concentrations, SHAP values of normalised 3-day mean u and v wind components were added up and transformed to degrees (Fig. 7). Generally, wind direction has a positive contribution

**Figure 6.** As Fig. 5 for MLH SHAP values (contribution of MLH to the prediction of species and total PM1 for each data instance) vs. absolute MLH values [m agl].

to the model outcome when winds from the northern to northeastern sectors prevail, while negative contributions are evident for southwesterly directions. Given the location of the measurement site, this pattern undoubtedly reflects the advection of particles emitted from continental Europe and/or Paris city centre under high pressure system conditions versus cleaner marine air masses during southwesterly flow (Bressi et al., 2013; Petetin et al., 2014; Petit et al., 2015; Srivastava et al., 2018).



Increased concentrations of organic matter are predicted for northerly, northeasterly and easterly winds. These patterns suggest

a significant contribution of advected organic particles from a specific wind sector. This is in agreement with the findings of Petetin et al. (2014) who estimated that 69 % of the PM25 organic matter fraction is advected by northeasterly winds, which is related to advected particles from wood burning sources in the Paris area and SOA formation along the transport trajectories. While Petit et al. (2015) did not find a wind direction dependence of organic matter measured at SIRTA using wind regression, they reported the regional background of organic matter to be of importance. Comparing upwind rural stations to urban sites,

Bressi et al. (2013) concluded organic matter is largely driven by mid- to long-range transport. Contributions to the SO4-model are highest for northeastern and eastern wind direction, which aligns with previous findings by Pay et al. (2012); Bressi et al. (2014); Petit et al. (2017), who identified the Benelux region and western Germany as strong emitters of sulfur dioxide (SO2). SO2 can be transformed to particulate SO4 (Pay et al., 2012) while being transported towards the measurement site. Nitrate concentrations are affected by long-range transport from continental Europe (Benelux, western Germany), which are

imported towards SIRTA from northeastern directions (Petetin et al., 2014; Petit et al., 2014). It is to be expected that the influence of mid- to long-range transport on the particle observations at SIRTA is rather substantial, with most high pollution days affected by particle advection from continental Europe (Bressi et al., 2013). Concerning BCff and BCwb, model results suggest a dependence on wind direction during northwestern to northeastern inflow. Although BC concentrations are expected to be largely determined by local emissions (Bressi et al., 2013), e.g. from local residential areas, a substantial contribution

of imported particles from wood burning and traffic emissions from the Paris city centre (Laborde et al., 2013; Petetin et al., 2014) and continental sources is likely (Petetin et al., 2014).

### 4.2.4 Contribution of feature interactions

Strong pairwise interactive effects are found between MLH vs. time since last precipitation and MLH vs. maximum wind speed and shown in Figs 8a and 8b. SHAP interaction effects between MLH and time since last precipitation are most pronounced for

MLHs below ∼ 500 m agl (Fig. 8a). Interaction values are negative for low MLHs paired with time since last precipitation close to zero hours. With increasing time since last precipitation, interaction effects become positive, thus increasing the contribution of Tprec and MLH to the model outcome. An explanation of this pattern concerning underlying processes could be that due to the lack of precipitation, a higher number of particles is available in the atmosphere for accumulation, hence increasing the accumulation effect of a shallow MLH. In case of recent precipitation, the accumulation effect of a shallow MLH is weakened.

For higher MLHs, interactive effects with time since the last precipitation event are marginal. Interactive effects between MLH and wind speed are shown in Fig. 8b. Positive SHAP values for maximum wind speeds below ∼2 m/s reflect stable situations, favoring the accumulation of particles, whereas high wind speeds enhance the ventilation of particles (Sujatha et al., 2016). This can also be deduced from Fig. 8b, which shows increased SHAP values for low wind speeds in combination with a low MLH. Low wind speeds combined with a high MLH (>∼1000 m agl), on the other hand, result in decreased SHAP values.

Similarly, low MLHs combined with higher wind speeds (>∼2 m/s) also decrease predictions of total PM1 concentrations. Maximum wind speed and time since last precipitation (plot not shown here) interact in a similar way. The positive effect of low wind speeds on the model outcome is increasing with increasing time since last precipitation.



**Figure 7.** As Fig. 5 for wind direction SHAP values (contribution of 3-day mean wind direction to the prediction of species and total PM1 for each data instance) vs. absolute wind direction [°].

### 4.3 Meteorological conditions of high-pollution events

To further identify conditions that favor high pollution episodes, the data set is split into situations with exceptionally high total
PM1 concentrations (>95th percentile) and situations with typical concentrations of total PM1 (interquartile range, IQR). This





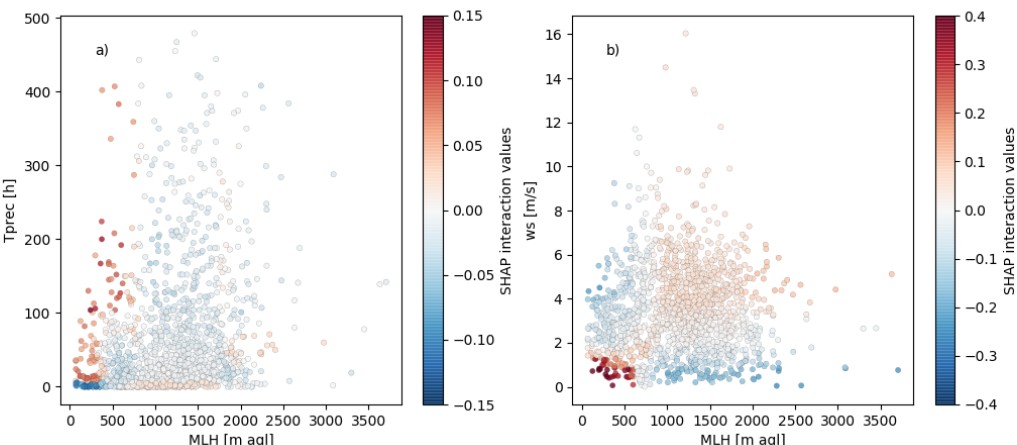

**Figure 8.** MLH vs. a) time since last precipitation and MLH vs. b) maximum wind speed, respectively, colored by the SHAP interaction values for the respective features.

is done for the meteorological summer and winter seasons to contrast dominant drivers between these seasons. Mean SHAP values refer to the total PM1 model, corresponding input feature distributions and species fractions for the two subgroups are aggregated seasonally. This allows for a quantification of seasonal feature contributions to average or polluted situations. Figs 9 & 10 show mean SHAP values for typical (left) and high-pollution (right) situations in the upper panel. The distribution of

SHAP values are shown as box plots for each feature. Absolute feature value distributions are given in the bottom of the figure. In the lowest subpanel, the chemical composition of the total PM1 concentration for each subgroup is shown. The largest contributor to high pollution situations in winter is air temperature (Fig. 9). SHAP values for temperature are substantially increased during high pollution situations, when temperatures are systematically lower. Further contributing factors to high pollution situations are the lows MLHs, low wind speeds, a high average NO3 fraction of the previous day and negative u (i.e.,

winds from the east) and v (i.e., winds from the north) wind components. In winter, the PM1 composition shows a relatively large fraction of nitrates, which is increased during high pollution situations (Fig. 9, lower panel). High concentrations of nitrate in winter can be linked to advection or to enhanced formation due to the temperature-dependent low volatility of ammonium nitrate (Petetin et al., 2014). The organic matter fraction is slightly decreased during high pollution situations. MLH and maximum wind speed contributions to high pollution situations are linked to low ventilation conditions which are

very frequent in winter ((Dupont et al., 2016). Positive contributions of wind direction for inflow from the northern and eastern sectors are dominant during high pollution situations while inflow from the southern and western sectors prevails during average pollution situations (see Fig. 7, Bressi et al., 2013; Petetin et al., 2014; Srivastava et al., 2018). Note that the time since the last precipitation is increased during high pollution situations, but the effects on the model outcome is weak. This suggests that lacking precipitation is not a direct driver of modelled total PM1 concentrations, but increases the contribution of other input

features (see Fig. 8a) or is a meaningful factor in only some situations.



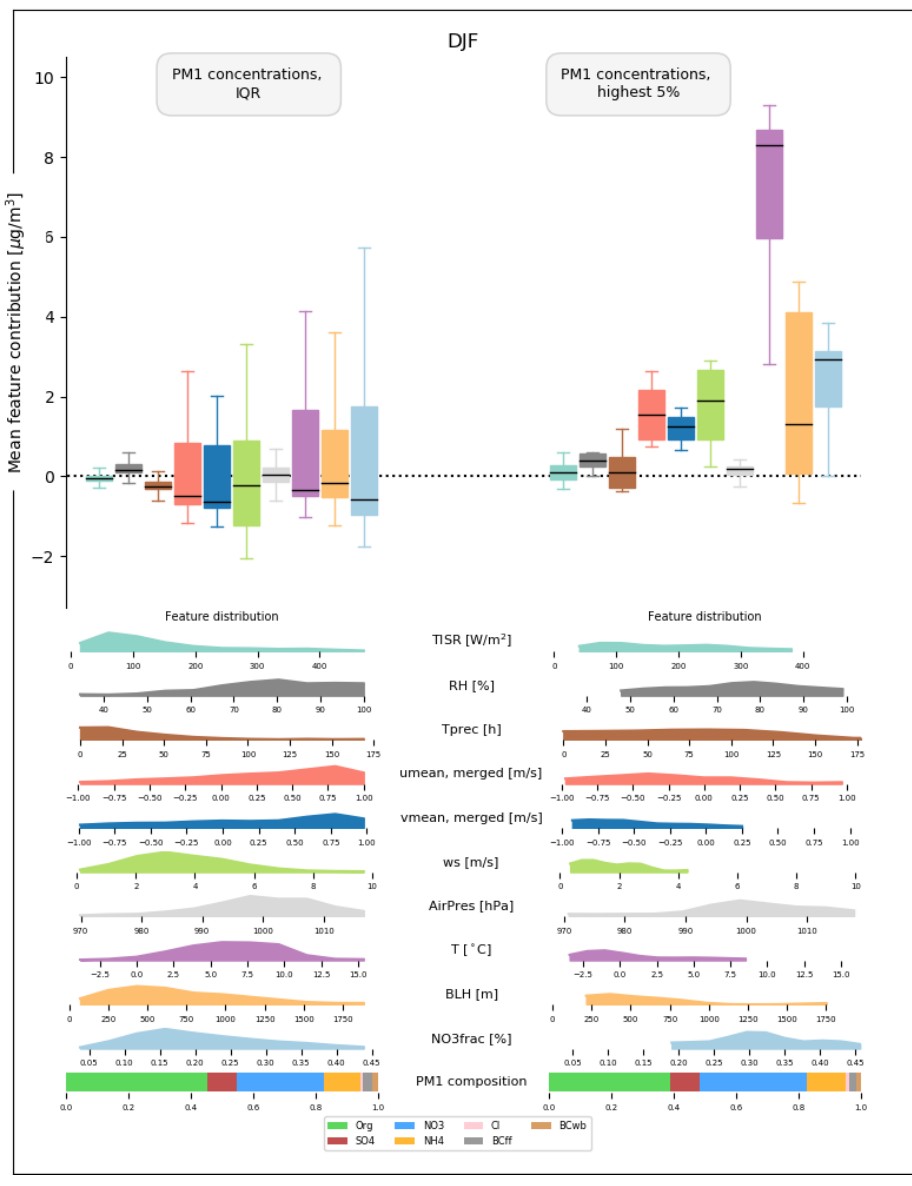

**Figure 9.** Mean feature contributions (i.e., SHAP values) for situations with low total PM1 concentrations (left) and situations with high pollution (right), respectively, during winter (December, January, February). Respective range of SHAP values by species are shown as box plots, with median (bold line), 25-75th percentile range (boxes), and 10–90th percentile range (whiskers). Both training and test data are included. Absolute feature value distributions (given as normalised frequencies) as well as the chemical composition of the total PM1 concentration are shown in the subpanels. Colors of the box plots correspond to colors in the feature distribution subpanels. SHAP values of the input features u_norm_3d and u_norm as well as v_norm_3d and v_norm were merged to u_norm, merged and v_norm, merged to achieve better transparency.



Summer total PM1 composition (Fig. 10) is characterised by a larger fraction of organics compared to the winter season (Fig. 9). As a considerable fraction of organic matter is formed locally (Petetin et al., 2014), the increased proportion of organics could be due to more frequent stagnant synoptic situations that may limit the advection of transported SIA particles. In addition, the positive SHAP values of solar irradiation and temperature highlight that the solar irradiation stimulates transformation pro-

cesses and increases the number of biogenic SOA particles (Guenther et al., 1993; Petetin et al., 2014). As mean temperatures are highest in summer, positive temperature SHAP values are associated with increased organic matter concentrations (Fig. 5). The higher importance (i.e. higher SHAP values) of time since the last precipitation event during high pollution situations points to an accumulation of particles in the atmosphere. Dry situations can also enhance the emission of dust over dry soils (Hoffmann and Funk, 2015). The negative contribution of MLH during both typical and high pollution situations reflects sea-

sonality, as afternoon MLHs in summer are usually too high to have a substantial positive impact on total PM1 concentrations (see Fig. 6). MLH is thus not expected to be a driver of day-to-day variations of summer total PM1 concentrations. Note that the average MLH is higher during high pollution situations, which likely points to increased formation of SO4 (see Fig. 6).

### 4.4 Day-to-day variability of selected pollution events

Analysing the combination of SHAP values of the various input features on a daily basis allows for direct attribution of

the respective implications for modelled total PM1 concentrations (Lundberg et al., 2020). Here, four particular pollution episodes are selected to analyse the model outcome with respect to physical processes (Figs 11-14). The examples highlight the advantages but also the limitations of the interpretation of the statistical model results. The high pollution episodes took place in winter 2016 (10th - 30th January and 25th November - 25th December), spring 2015 (11th - 31st March) and summer 2017 (8th - 28th June). The upper panels in Figs 13-16 indicate the total PM1 prediction as horizontal black line with vertical black

lines denoting the range of predictions of all 10 models. The observed species concentrations are shown in the corresponding colors. The subsequent panels show absolute values and SHAP values for the most relevant meteorological input features.

### 4.4.1 January 2016

Prior to the onset of the high-pollution episode in January 2016 (Fig. 11), the situation is characterised by MLHs in the range of 1000m, temperatures above freezing ($\sim$5-10°C), frequent precipitation and winds from the southwest. The organic matter

fraction dominates the particle speciation. The episode itself is reproduced well by the model. According to the model results, the event is largely temperature-driven, i.e. SHAP values of temperature explain a large fraction of the total PM1 concentration variation (note the adjusted y-axis of the temperature SHAP values). On 18th January, temperatures drop below freezing, coupled with a decrease in MLH. As a consequence, both modelled and observed PM1 concentrations start to rise. A further increase in total PM1 concentrations is driven by a sharp transition from stronger southwestern to weaker northeastern winds

(strong negative u component, weak negative v component) on January 19th. The combined effects of these changes lead to a marked increase in total PM1 concentrations, peaking at $\sim$37 $\mu$g/m$^3$ on 20th January. On the following days, temperatures increase steadily, thus the contribution of temperature decreases. At the same time, although values of MLH remain almost constant, the contribution of MLH drops substantially from $\sim$5 $\mu$g/m$^3$ to $\sim$2 $\mu$g/m$^3$. This is due to interactive effects between

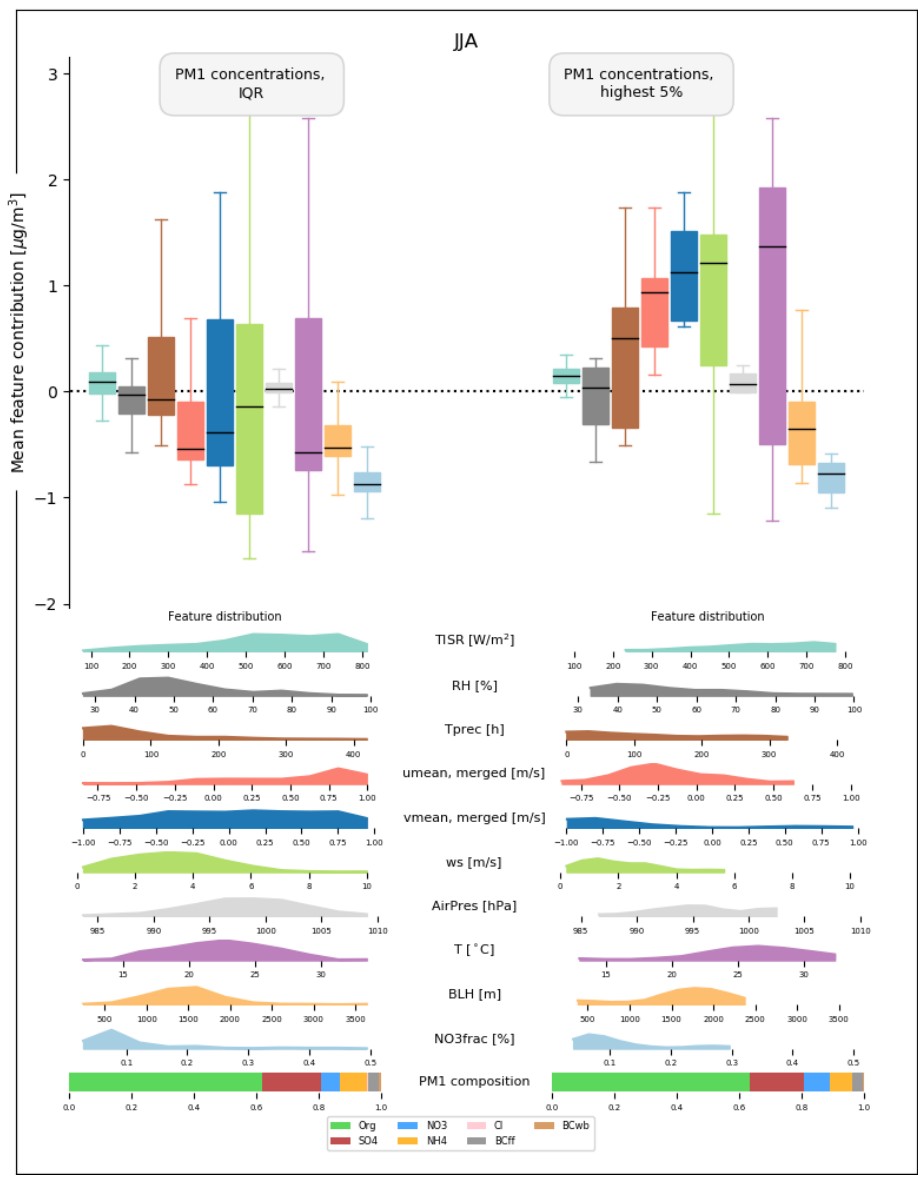

**Figure 10.** As Fig. 9 for mean feature contributions (i.e., SHAP values) for situations with low total PM1 concentrations (left) and situations with high pollution (right), respectively, during summer (July, June, August).

MLH and the features wind speed, time since last precipitation and normalised v-wind-component. All of these features in-

crease the contribution of MLH on 20th January, but decrease its contribution on 21st-23rd January. The high pollution episode ceases after a shift to southeastern winds and the increasing temperatures. The pollution episode is characterised by a relatively large fraction of NO3 and NH4, which explains the strong feature contribution of temperature to the modeled total PM1 concentration, as the abundance of these species is temperature dependent (see Fig. 5) and points to a large contribution of locally





formed inorganic particles. Still, the contribution of wind direction and speed also suggests that advected secondary particles
and their build-up in the boundary layer are relevant factors during the development of the high pollution episode (Petetin et al.,
2014; Petit et al., 2014; Srivastava et al., 2018).

### 4.4.2   December 2016

A high-pollution episode with several peaks of total PM1 is observed in November and December 2016. The first peak on
26th December is followed by an abrupt minimum in total PM1 concentrations on 28th November, and a build-up of pollution
in a shallow boundary layer towards the second peak on 2nd December with total PM1 concentrations exceeding $40 \, \mu g/m^3$.
In the following days, total PM1 concentrations continuously decrease, eventually reaching a second minimum on 11th De-
cember. A gradual increase in total PM1 concentrations follows, resulting in a third (double-)peak total PM1 concentration
on 17th December. Total PM1 concentrations drop to lower levels afterwards. Throughout the 3.5 week-long episode, high
pollution is largely driven by shallow MLH ($<\sim500m$), and weak north-northeasterly winds, i.e. a regime of low ventilation
associated with high pressure conditions favorable for emission accumulation. During the brief periods with lower total PM1
concentrations, these conditions are disrupted by a higher MLH ($\sim$28th November), or a change in prevailing winds ($\sim$11th
December). In contrast to the pollution episode in January 2016, this December 2016 episode is not driven by temperature
changes. Temperatures range between $\sim$5-12°C and have a minor contribution to predicted total PM1 concentrations (see also
Fig. 5), emphasizing the different processes causing air pollution in the Paris region. Note that the model is not able to fully re-
produce the pollution peak on December 2nd, which may be indicative of missing input features in the model. Judging from the
PM1 species composition during this time (relatively high fraction of NO3 and BC), it seems likely that missing information
on particle emissions may be the reason for the difference between modeled and observed total PM1 concentration.

### 4.4.3   June 2017

A period of above average total PM1 concentrations occurred in June 2017. The episode is very well reproduced by the model,
suggesting a strong dependence of the observed total PM1 concentration to meteorological drivers. Although absolute total
PM1 concentrations are substantially lower than during the previously described winter pollution episodes, the event is still
above average for summer pollution levels. Organic matter particles dominate the PM1 fraction throughout the episode, with
a relatively high SO4 fraction. Conditions during this episode are characterised by strong solar irradiation (positive SHAP
values) and high MLHs (mostly negative SHAP values), which show low day-to-day variability and reflect characteristic
summer conditions. A lack of precipitation (no rain for a period of more than 2 weeks) and high temperatures also contribute
to the total PM1 concentrations during this episode. While solar irradiation and time since last precipitation are associated
with positive SHAP values throughout this period, air temperature only has a positive contribution when exceeding $\sim$25 °C.
This aligns with patterns shown in Fig. 5, where increased concentrations of organic matter and SO4 are identified for high
temperatures. Peak total PM1 concentrations of $\sim$17 $\mu g/m^3$ are observed on June 20th and 21st. A change in the east-west
wind component from western to eastern inflow directions in conjunction with an increase in temperatures to above 30 °C
are the drivers of the modeled peak in total PM1 concentrations. MLH is also increased with values $\sim$2000 m agl, which are

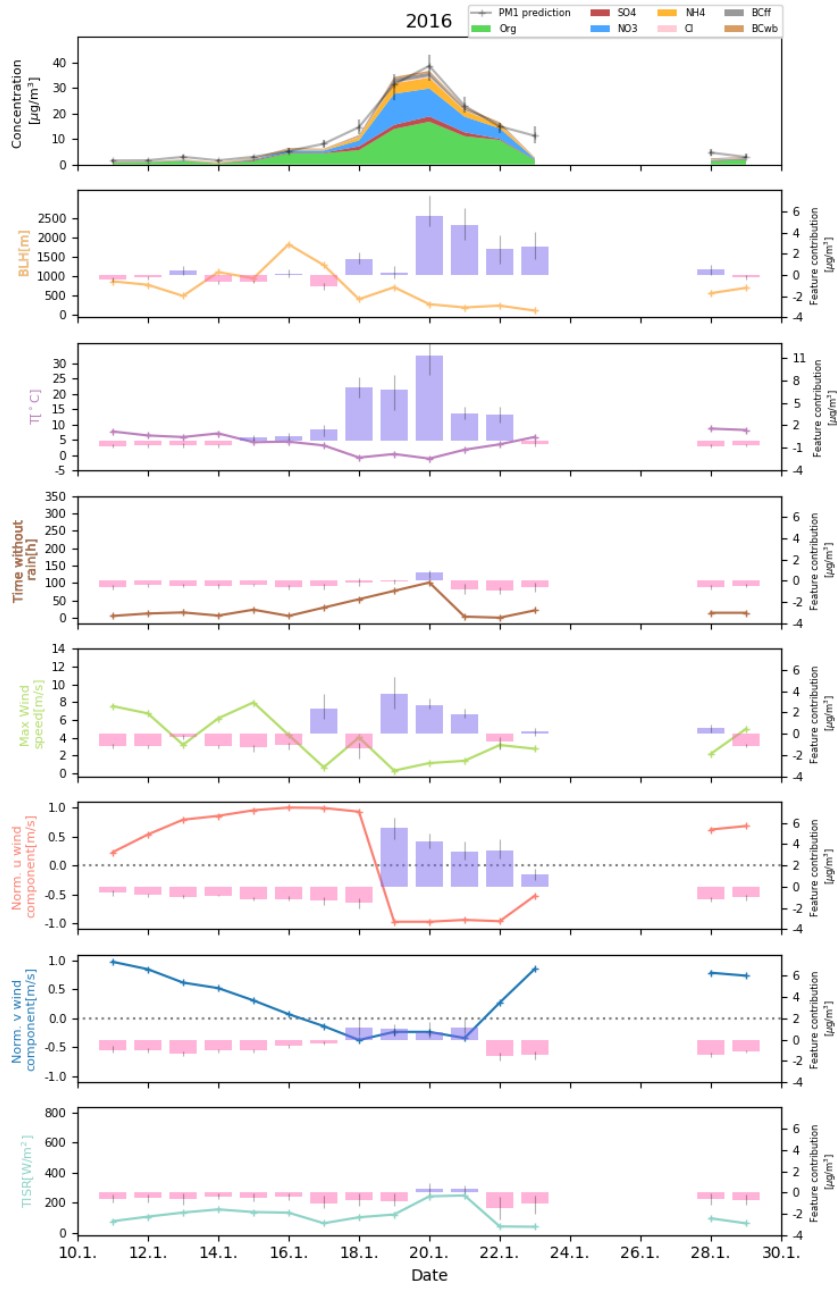

**Figure 11.** Winter pollution episode in January 2016. (a): predicted total PM1 and observed PM1 species concentrations, with absolute input feature values and corresponding SHAP values of (b) MLH, (c) temperature, (d) hours after rain, (e) maximum wind speed (f) normalised u wind and (g) normalised v wind component.





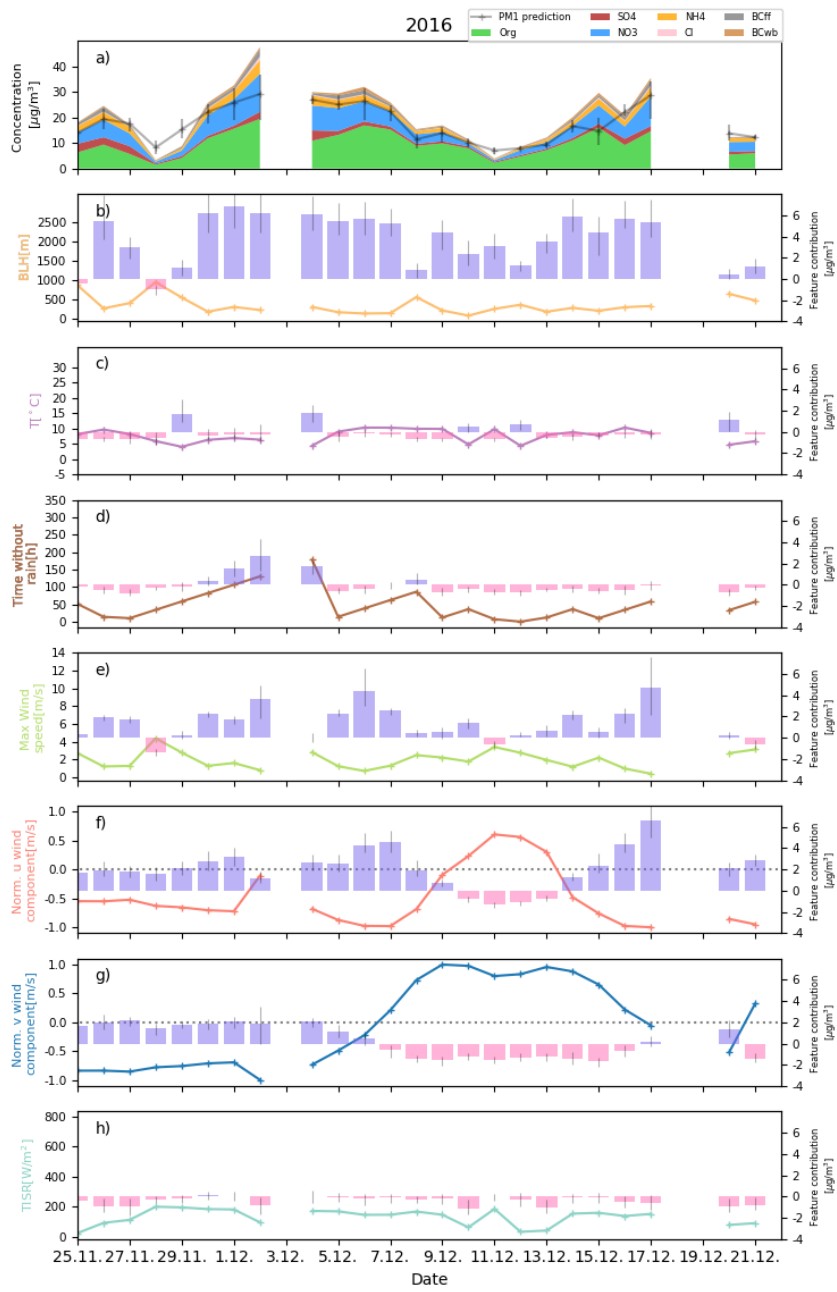

**Figure 12.** As Fig. 11 for a further winter pollution episode in December 2016.

associated with slightly positive SHAP values. This observation fits with findings described in section 4.2.2 and is likely linked to enhanced secondary particle formation (Megaritis et al., 2014; Jiang et al., 2019). As suggested by response patterns of species to changes in MLH shown in Fig. 7, this effect is linked to an increase in SO4 concentrations. The main fraction of the



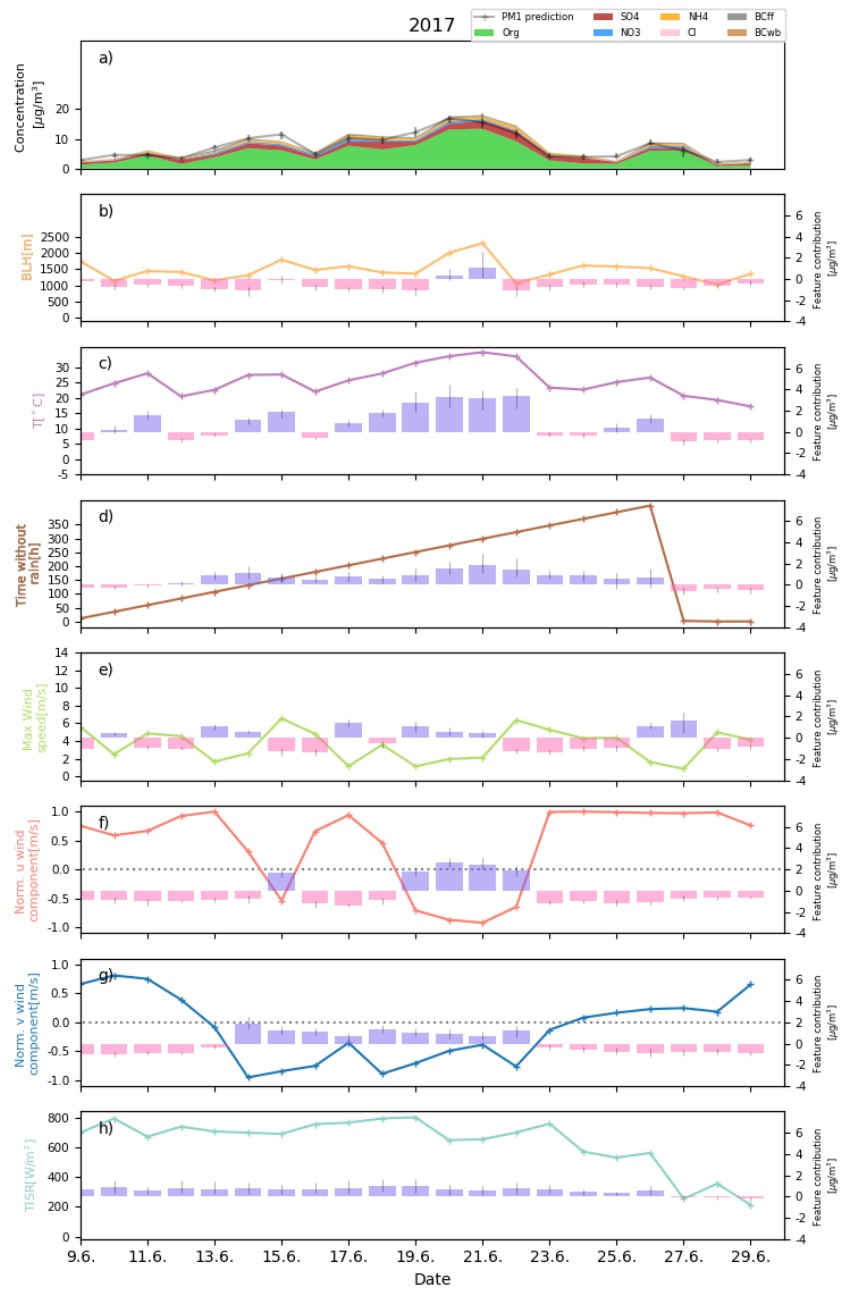

**Figure 13.** As Fig. 11 for an exemplary summer pollution episode in June 2017.

peak total PM1 values, however, is linked to an increase in organic matter concentrations due to the warm temperatures (see Fig. 5).





### 4.4.4 March 2015

High particle concentrations are measured in early March 2015 with high day-to-day variability. This modelled course of the pollution episode is chosen to compare results to previous studies focusing on the evolution of this episode (Petit et al., 2017; Srivastava et al., 2018). The episode is characterised by high fractions of SIA particles, in particular SO4, NH4 and NO3 (Fig. 14, upper panel) and similar concentrations observed at multiple measurement sites in France (Petit et al., 2017). Contributions of local sources are low and much of the episode is characterised by winds blowing in from the northwest, advecting aged SIA particles (Petit et al., 2017; Srivastava et al., 2018) and organic particles of secondary origin (Srivastava et al., 2019) towards SIRTA. A widespread scarcity of rain probably enhanced the large-scale formation of secondary pollution across western Europe (in particular western Germany, The Netherlands, Luxemburg, Petit et al., 2017), which were then transported towards SIRTA. This is reflected by the SHAP values of the u and v wind components, which are positive throughout the episode (see Fig. 14g & 14h). Concentration peaks of total PM1 are measured on 18th and 20th March. Both peaks are characterised by a rapid development of total PM1 concentrations. As described in Petit et al. (2017), these strong daily variations of total PM1, which are mainly driven by the SIA fraction, could be due to varying synoptic cycles, especially the passage of cold fronts. The contribution of MLH and temperature is relatively small, which is consistent with the high influence of advection on total PM1 concentrations during the episode. The exceptional character of the episode (see Petit et al., 2017) partly explains the bad performance of the model in capturing total PM1 variability during the event. Unusual rain shortage is observed in large areas of Western Europe prior to the episode (Petit et al., 2017). While time since precipitation at the SIRTA-site is a large positive contributor to the model outcome (see Fig. 14d), it is not driving the day-to-day variations. The unusual nature of this event and lacking information on emission in the source regions and formation processes along air mass trajectories in the model likely explain why the model has difficulties in reproducing this pollution episode. While this has implications for the application of explainable machine learning models for rare events, this is not expected to be an issue for the other cases and seasonal results presented here.

## 5 Conclusions and outlook

In this study, dominant patterns of drivers of PM1 species and total PM1 concentrations are identified and analysed using a novel, data-driven statistical approach. A machine learning model is set up to explain measured speciated and total PM1 concentrations based on meteorological measurements from the SIRTA supersite, southwest of Paris. The statistical model is able to reproduce daily variability of particle concentrations well, and is used to analyse and quantify the atmospheric processes causing high-pollution episodes during different seasons using a SHAP-value framework. Comparison of the results based on the statistical model with findings of previous studies on air quality patterns in the Paris region shows good agreement. Peak concentrations of total PM1 in winter are mainly driven by shallow MLHs, low temperatures and low wind speeds. These conditions are often amplified by northeastern wind inflow, i.e. high-pressure conditions. A detailed analysis revealed different drivers of high-pollution episodes in winter. For an episode in January 2016, model results show a strong contribution of temperature to the elevated PM1 concentrations during this episode (up to 11 $\mu$g/m$^3$ are attributed to temperature), suggesting

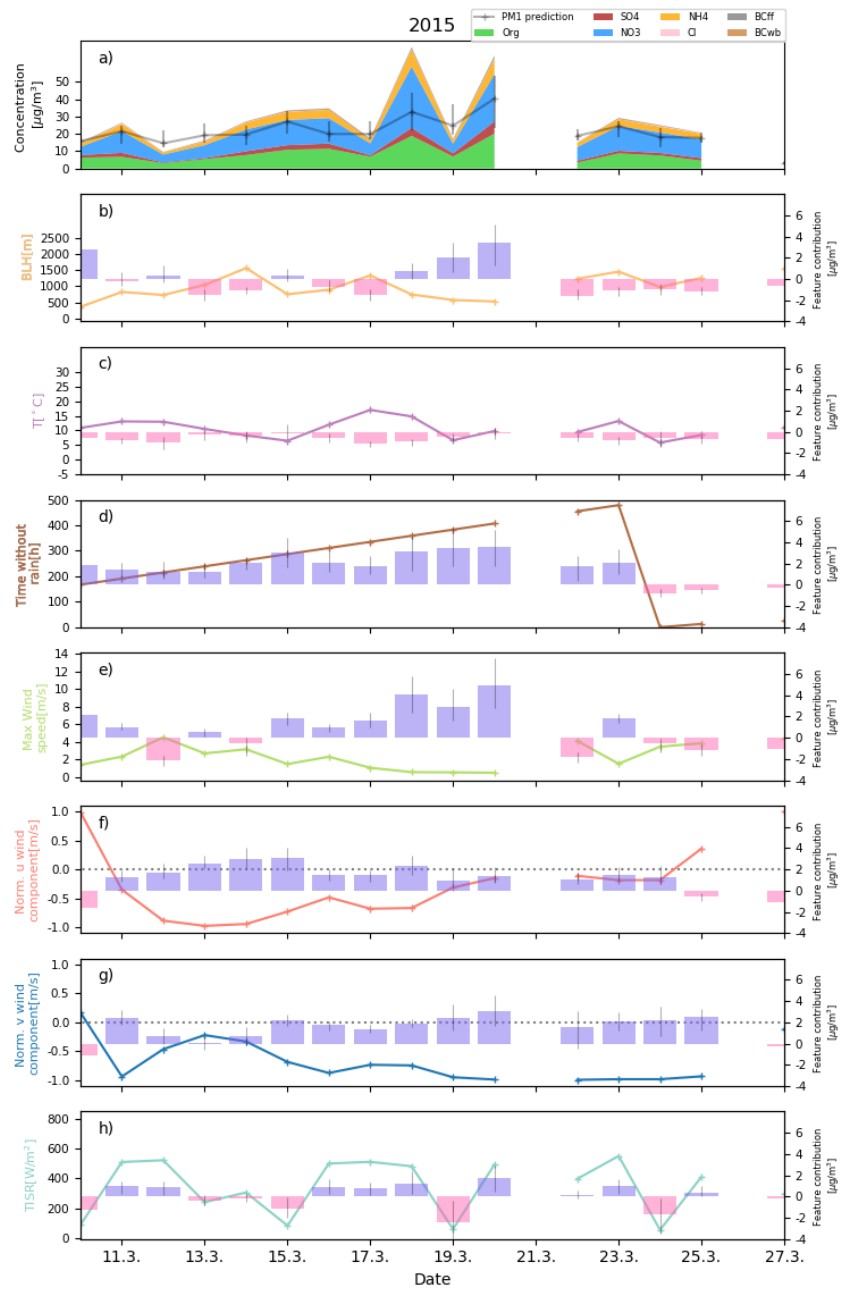

**Figure 14.** As Fig. 11 for an exemplary spring pollution episode in March 2015.

enhanced local, temperature-dependent particle formation. During a different, prolonged pollution episode in December 2016, temperature levels were relatively stable and had no influence. Here, contributions of MLHs (<500 m asl) were quantified to be the main driver of PM1 peak concentrations with contributions up to 6 $\mu$g/m$^3$, along with wind direction contributions of up to





$\sim 6\,\mu\text{g/m}^3$. Total PM1 concentrations in spring can be as high as $50\,\mu\text{g/m}^3$. These peaks in spring are not as well reproduced by the model as winter episodes and are likely due to new particle formation processes along the air mass trajectories, in

particular of nitrate. Summer PM1 concentrations are lower than in other seasons. Model results suggest that summer peak concentrations are largely driven by high temperatures, particle advection from Paris and continental Europe with low wind speeds and prolonged periods without precipitation. For an example episode in June 2017, temperatures above $30°\text{C}$ contribute $\sim 3\,\mu\text{g/m}^3$ to the total PM1 concentration. On site scarcity of rain increases air pollution, but does not appear to be a main driver of strong day-to-day variations in particle concentrations. Presumably, this is because droughts are synoptic and are spread over

several days or even weeks. Thus, they present very low inter-daily variability on the local scale. Nonetheless, Petit et al. (2017) have highlighted the link between extreme PM concentrations (especially during spring) and extreme precipitation deficit (compared to average conditions). Main drivers of day-to-day variability of predicted PM1 concentrations are changes in wind direction, air temperature and MLH. These changes often superimpose the influence of time without precipitation. Individual PM1 species are shown to respond differently to changes in temperature. While SO4 and organic matter concentrations are

increased during both high and low temperature situations, NH4 and NO3 are substantially increased only at low temperatures. Model results indicate that SIA particle formation is enhanced during shallow MLH conditions.

Many of the results presented here hold true for regions other than suburban Paris and are thus beneficial for the general understanding of drivers of air pollution. This includes the importance of formation processes of secondary pollutants as well as the dominant role of the MLH for PM1 concentrations. The contribution of wind direction furthermore highlights the role of

advected pollution and emphasizes the need for large-scale measures against air pollution. The GBRT approach in combination with the SHAP regression values presented here provides an intuitive tool to assess meteorological drivers of air pollution and to advance the understanding of high pollution events by uncovering different physical mechanisms leading to high-pollution episodes. To our knowledge, this is the first time that the SHAP-framework for explainable machine learning is applied in atmospheric sciences. The results of this study are highly relevant for policy makers, e.g. by providing a basis for future clean

air programs or by providing the potential of a statistically-based early warning system for high pollution episodes. Statistical models such as the one developed in the present study can be used to attribute changes in air quality to policy measures (e.g., Cermak and Knutti, 2009), and may help political decision makers develop and implement effective clean-air policies. Future efforts could also combine the statistical model framework with short-term weather forecasts, which would allow to provide an air quality forecast based on the predictions of the statistical models, taking into account expected meteorological conditions.

This study could be extended in the future, e.g. by including information on anthropogenic emissions or further stations down- and upwind of SIRTA, which would allow further analysis of dominant advection patterns. Furthermore, information on emissions or meteorology in the source region of air masses e.g., using satellite-based observations, might be helpful to better reproduce particle transport patterns. This could be complemented by incorporating synoptic variables, e.g., the North Atlantic Oscillation (NAO) index.





*Author contributions.* Conceptualization, R.S., J.C., M.H., S.K.; Data acquisition, J.-E.P., O.F., M.H., and S.K.; Formal analysis, R.S; Investigation, R.S.; Methodology, R.S., J.C., M.H., S.K, H.A., J.F. and M.K.; Visualization, R.S.; Writing-original draft, R.S.; Writing-review & editing, R.S., J.C., M.H., S.K, J.F., H.A., J.-E.P., O.F., and M.K.

*Competing interests.* The authors declare that they have no conflict of interest.

*Acknowledgements.* The authors would like to acknowledge the ACTRIS-2 project that received funding from the European Union's Horizon 2020 research and innovation programme under grant agreement No 654109. Acknowledgements are extended to Rodrigo Guzman and Christophe Boitel for providing the latest update of the SIRTA ReOBS dataset. Furthermore, the authors acknowledge Scott Lundberg for his work on the TreeSHAP algorithm. R.S. was supported by the KIT Graduate School for Climate and Environment (GRACE).



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
