# Peer review of "Meteorology-driven variability of air pollution (PM1) revealed with explainable machine learning"

_Atmospheric Chemistry and Physics, 2020_

## Editor Comment (EC1) · Leiming Zhang (Editor) · 4 Aug 2020

Comments previously provided by a reviewer are listed below. These comments have not been addressed in this ACPD version and thus need to be addressed in this round of review.

I am afraid that even after reading the article, I do not understand why the authors choose to explain air quality over Paris based on meteorology at the SIRTA location, when regional and local emissions and atmospheric transformations during long range transport are the major drivers of ambient pollution. These major drivers are mentioned towards the end as future work, but studies should start there. For example, even if

MLH or wind speed is low, zero emissions = no air pollution.

The authors frame it as "we should take atmospheric and environmental processes into account during the development of efficient pollution mitigation strategies"/"a basis for future clean air programs", but AirParif can't exactly change wind conditions or MLH or T/RH. Maybe this can be used to forecast periods of bad air quality - but they describe some important events that the model fails to reproduce because it is missing major drivers in the inputs (lines 390, 426-427). So I am not sure this study is an advance over previous knowledge.

---

## Referee Comment (RC2) · Anonymous Referee #3 · 16 Sep 2020

This paper presents a machine-learning built model approach to analyse an extensive multi-parameter dataset at observational in a suburban area south of Paris. The focus of the manuscript is using a recently published tool ("SHapley Additive exPlanation (SHAP) values") to analyse the machine-learning model's predictions and then attribute drives of the statistical model.

The paper presents large amounts of information about the output from the analysis tool, but not enough focused justification or evidence is presented about how novel these interpretations are or how that they could be used for air pollution mitigation policy etc. At points, the paper even reads as if the authors are suggesting that authorities

seek to mitigation against the meteorology contribution to air pollution. Could this analysis be used to make a forecasting tool if parameters were gained in real-time? If so, how long ahead would these predictions be expected to be useful for? Would this be useful in a public health context?

A core premise (in the abstract and elsewhere) is that we do not fully understand the contribution of meteorology to high air pollution episodes is true, however, this does justify the framework used here which omits two other key drivers (chemistry and emissions). Apart from a few mentions, it is not clear how are these contributions and considered in this method. Are the contributions of these processes just assumed to be part of the meteorological contributions? This needs to be a lot clearer.

The paper seems mostly focused on exploring the "SHapley Additive exPlanation (SHAP) values" approach and it is unclear whether a novel contribution has been made to the field of air pollution research. This paper may be better suited to a machine learning journal or could be re-write to be more focused on air pollution. Either of these two options would require large changes to the current manuscript.

Specific comments

Why has PM1 been the focus of this study, rather than the more health-relevant PM2.5 species? Also, how did the model perform at predicting PM10? Considering the omission of chemistry and emissions in this study, would PM10 or PM2.5 be a better candidate for study?

Line 21 - "Processes vary even within seasons"

This does not read well. Of course, processes will vary within seasons.

Line 24 - "likely causes an increase in local wood-burning emissions"

Cause and effect seem to be muddled. Maybe the authors mean to say increases in burning emission could explain increased particulates?

line 25 - "The application of SHAP regression values within a machine learning framework presents a novel and promising way of analysing observational data sets in environmental sciences."

Are there implications for what we should focus on meteorology studies or observations on? What about the implications for air-quality modelling or policy? Just presenting another tool that can be used is not a notable contribution.

Line 90 - How can policymakers use this information? Improve air quality models? Focus research directions? What about it is new?

Line 90 - Why not focus on the SIRTA region, rather than Paris, which is in completely different chemistry and emissions regime? The reader needs to be convinced that the site is representative of the Paris region.

Technical comments

Please use sub/superscripts for chemical species throughout (e.g. $SO_4^{2-}$, $SO_2$, $PM_{2.5}$).

Expand acronyms in sub-header titles (e.g. MLH).

Expand acronyms once per major section too.

---

## Author Response (AR1)

**Rebuttal ACPD**

**BOLD** = reviewer comment
*Italic* = answer to reviewer comment
Red = highlighted changes in manuscript

**Comments previously provided by a reviewer**

(Editor: These comments have not been addressed in this ACPD version and thus need to be addressed in this round of review.)

**I am afraid that even after reading the article, I do not understand why the authors choose to explain air quality over Paris based on meteorology at the SIRTA location, when regional and local emissions and atmospheric transformations during long range transport are the major drivers of ambient pollution. These major drivers are mentioned towards the end as future work, but studies should start there. For example, even if MLH or wind speed is low, zero emissions = no air pollution.**

*Answer:*
*Thank you for this comment.*
*Indeed we do not inted to explain Paris air quality using the SIRTA site - the focus is on air quality at the SIRTA site, which is near Paris and representative of suburban background concentrations. We have clarified this in the manuscript (see changes in L11, L88, L94, L127, L234, L502). The aim of this study is to quantify how meteorological factors influence pollutant concentrations and thus add to system understanding. It has been shown in previous studies that pollutant concentrations are not solely driven by emissions, and can be exacerbated or weakened substantially by certain meteorological conditions. For SIRTA, this has been described for example by Dupont et al., 2016.*
*It was not intended to set up a prognostic model to forecast PM1 as accurately as possible in time.*
*Emissions of pollutants or precursor gases undoubtedly constitute a prerequisite to air pollution, but pollutant concentrations are not solely driven by anthropogenic emissions, but strongly affected by varying amounts of natural background emissions (see e.g., Liora et al., 2016, DOI: 10.1016/j.atmosenv.2016.04.040, Jiang et al. 2019, DOI: 10.5194/acp-19-15247-2019). Thus, high concentrations of particulates could also occur*

*during episodes with low anthropogenic emissions. This is particularly the case in summer, when biogenic organic emissions are high (this is mentioned in the introduction, L56).*

*The influence of meteorology can lead to quite different air pollution situations, even if emissions are constant. In winter, meteorological conditions exert great influence on formation pathways, as we describe in chapter 4.2.1. For example, condensation of ammonium nitrate in the aerosol phase is enhanced at low ambient temperatures and high relative humidity  (see e.g., Pay et al., 2012; Bressi et al., 2013; Petetin et al., 2014; Petit et al., 2015). Hence, even if emissions would be not above average levels, this formation mechanism would increase the concentrations of pollutants.*

*Transformation processes are partly covered by meteorological parameters, e.g., through the influence of temperature (please see also answers to Referee #3 comments). Obviously, the model does still not capture all of the occurring variance of PM1 concentrations, but since it was not the intention to set up a predictive framework, the focus is not primarily on accuracy, but on interpretability. Hence, the included parameters are deemed adequate for the analysis*

*In the updated version of the manuscript, we have taken great care to consider this comment and now more clearly communicate the main goal of this manuscript:*

- *L3: However, the scientific understanding of the ways by which complex interactions of meteorological factors lead to high pollution episodes is inconclusive, as the effects of meteorological variables are not easy to separate and quantify*
- *L6: In this study, a novel, data-driven approach based on empirical relationships is used to characterise, quantify and better understand the meteorology-driven component of PM1 variability.*
- *L8: Changed to "Based on the model, an isolation and quantification of individual meteorological influences for process understanding is achieved using SHapley Additive exPlanation (SHAP) regression values.*
- *L87: Changed to "Here, the multivariate and highly interconnected nature of the processes determining local PM1 concentrations is analysed in a data-driven way. Therefore, a state-of-the-art explainable machine learning model is set up to reproduce the variability of PM1 concentrations, thereby capturing empirical relationships between PM1 concentrations and meteorological parameters. The goal is to separate and quantify influences of the meteorological variables on PM1 concentrations to advance the process understanding of the complex mechanisms that govern pollution concentrations at the measurement site."*

**The authors frame it as "we should take atmospheric and environmental processes into account during the development of efficient pollution**

**mitigation strategies"/"a basis for future clean air programs", but AirParif can't exactly change wind conditions or MLH orT/RH.**

*These statements were aimed to show the potential benefits of considering atmospheric and environmental conditions when future measures to prevent air pollution are discussed. This relates mainly to three points:*

- *a) A realistic assessment of the effectiveness of measures against air pollution needs to take atmospheric and environmental processes into account as these processes partly control its variability. For example, if changes in PM concentrations due to traffic restrictions were to be determined, a simple comparison of pre-restriction and post-restriction concentrations would not be sufficient, as meteorological influences would be omitted. Machine learning approaches can be very useful to characterize the efficiency of mitigation policies. Recent lockdown in Spain is an adequate example here (Petetin et al., 2020, DOI: https://doi.org/10.5194/acp-20-11119-2020)*
- *b) Weather conditions which exacerbate pollutant concentrations are identified using the SHAP framework. On this basis, air-pollution measures could be adjusted depending on expected meteorological conditions. For example, warnings could be expressed to the public to remain vigilant or stay at home if possible.*
- *c) In a changing climate, more unfavorable meteorology could trigger and/or exacerbate PM pollution episodes, lowering the role of emission restrictions.*

*To make this clearer in the manuscript, the following changes were made:*
- *L47: The sentence "It is therefore crucial to take atmospheric and environmental processes into account during the development of efficient pollution mitigation strategies" was removed .*
- *The explanation relating to future clean air program was shifted to the end of the introduction and expanded; L96-106 now read:*
*"... allowing to infer meteorology-dependent processes driving PM concentrations at high temporal resolution.. Typical situations that lead to high PM1 concentrations are identified, serving as a decision support to policymakers to issue preventative warnings to the public if these situations are to be expected. In addition, by directly accounting for meteorological effects on PM1 concentrations, such a machine learning-based framework could help in assessing the effectiveness of measures towards better air quality. Furthermore, the proposed ML framework can be viewed as a first step towards a data-driven, prognostic tool in operational air quality forecasting, complementary to CTM approaches.*

**Maybe this can be used to forecast periods of bad air quality - but they describe some important events that the model fails to reproduce because it is missing major drivers in the inputs (lines 390, 426-427).**

*We show several examples where the model is well able to reproduce episodes of high pollutant concentrations (sections 4.4.1-4.4.3). This is encouraging and shows the appropriateness of the approach. There are of course also situations in which the model fails to reproduce high-pollution situations. These situations are shown in detail to stimulate further research in this direction.*
*As stated earlier, the current model setup was chosen for the purpose of improving the understanding of how meteorological factors influence pollutant concentrations, and to quantify potential influences (which was more clearly stated in L87-105). Setting up a probabilistic forecast model based on top of the presented framework would undoubtedly require many adjustments to include the factors correctly pointed out by the referee..*

**So I am not sure this study is an advance over previous knowledge.**
*Extensive changes were made to the manuscript to emphasize new scientific insights (see previous answers and answers to Referee#3 comments)*

**Anonymous Referee #1**

**The research work employs field measurements of Particulate Matter smaller than1μm in diameter (PM1), routine meteorology data and propose a machine learning framework in air-pollution forecasting. The authors address the significant challenge of Interpretability in machine learning using the SHapley Additive exPlanation (SHAP)regression values.**

**A general comment is related to the aim of the proposed work. Although the role of meteorology on PM concentrations is well studied, the paper proposes a novel method/tool of explainable machine learning in atmospheric sciences. The results support the use of explainable machine learning as a statistical modeling framework in operational air quality forecasting. The authors comment on this in the conclusion section but could highlight the ability of the proposed framework earlier in the manuscript.**

*Thank you for your assessment.*
*In order to highlight the capability of the framework in air quality forecasting, the*

*following statement was added to the introduction (L102): "Furthermore, the*
*proposed ML framework can be viewed as a first step towards a data-driven,*
*prognostic tool in operational air quality forecasting, complementary to CTM*
*approaches.."*

**While the results of the study are of local interest the proposed modelling framework has a high replication potential in areas with limited PM1 field measurements and therefore has a general implication in atmospheric science. Some concern is related to the use of meteorological data for the period of July to mid-November 2016. It is useful to include some descriptive statistical analysis of the meteorological data for all sites in order to compare and highlight the suitability of using meteorological data from the Paris Charles de Gaulle Airport.**
*Please note that only MLH was substituted during that time, all the other variables were not affected by the instrument failure.*
*An appendix was added. Figs A1 and A2 now provide a comparison of MLHs measured at Sirta vs. MLH measured at Charles de Gaulle airport for available data of the year 2016.*

**Furthermore, the locations of all measurement sites should be included in the map of the area of study and use more appropriate location mark labels.**
*Map (Figure 1) was changed accordingly.*

**The authors could provide some descriptive statistical analysis of the PM1 field mea-surements. This analysis could provide thresholds of high-pollution events in the region(e.g. similar to the >95 percentiles used in the paper).**
*Table showing mean, interquartile range, 95 percentile was added in section 4.3*
*Several thresholds were tested to define high-pollution events (e.g., >75%, >90%, >95%). It was found that the more extreme the events, the clearer the meteorological influence, which points to a relatively narrow set of meteorological characteristics responsible for high-pollution events. The decision to finally use the 95th percentile for the analysis was because this is a typical threshold for extreme value analysis. Below is shown the plot for extreme events >75th percentile; patterns are similar, but less distinct compared to >95th percentile as shown in the manuscript.*

[Figure]

Please note that a more extensive description of the statistics of the PM1 field measurements is also provided in Petit et al., 2014 (DOI: https://doi.org/10.5194/acp-14-13773-2014), Petit et al., 2015 (DOI: https://doi.org/10.5194/acp-15-2985-2015), Petit et al., 2017 (DOI: https://doi.org/10.1016/j.atmosenv.2017.02.012)

**The overall ability of the proposed framework could be also evaluated using exceedances forecast verification metrics (e.g. Probability of Detection, False Alarm Ratio etc.) for certain PM1 thresholds. This analysis could be complementary to the analysis of high-pollution case-studies and role of meteorological conditions of high-pollution events.**

*As the main objective of this study is to advance the understanding of meteorological drivers, a regression model was set up and validated to reproduce the temporal development of PM concentrations. The model that was set up for this task does not do a classification, therefore calculating the Probability of Detection or False Alarm Rate might not be suitable for validation. Hence, while these are certainly good suggestions for a classification framework, they are out of the scope of the manuscript and could be included in future work on this topic.*

**The paper presentation and structure is clear and supports the discussion of the results. The authors give proper credit to earlier published work and discuss their findings appropriately. The figures in the manuscript support the discussion of the results.**
*Thank you!*

**In Figures 9 and 10 the color-bar of the PM1 composition could be misleading. It is advised to change to avoid confusion with the feature contributions color scales.**
*The colors used for major PM1 species are consistent with previous literature (e.g., Petit et al. 2014, Petit et al. 2015, Dupont et al. 2016), so the colors of the meteorological variables were changed instead. In addition, a horizontal line was added to more clearly separate the PM1 composition plot.*

**The authors should also check of consistency of abbreviations throughout the manuscript. For example, Mixed-layer height (MLH) in some figures is abbreviated as BLH.**
*Thank you, BLH was changed to MLH in all affected plots.*

**Anonymous Referee #3**

**This paper presents a machine-learning built model approach to analyse an extensive multi-parameter dataset at observational in a suburban area south of Paris. The focus of the manuscript is using a recently published tool ("SHapley Additive exPlanation(SHAP) values") to analyse the machine-learning model's predictions and then attribute drives of the statistical model.**

**The paper presents large amounts of information about the output from the analysis tool, but not enough focused justification or evidence is presented**

**about how novel these interpretations are or how that they could be used for air pollution mitigation policy etc. At points, the paper even reads as if the authors are suggesting that authorities seek to mitigation against the meteorology contribution to air pollution. Could this analysis be used to make a forecasting tool if parameters were gained in real-time? If so,how long ahead would these predictions be expected to be useful for? Would this be useful in a public health context?**

*Thank you for your assessment.*

*The focus of this study is not on the prediction of pollutant concentrations in time, but to contribute to the advancement of the scientific understanding of how meteorology influences air pollution. The machine-learning framework presented in this study provides observation-based, quantitative estimations for the influence of various meteorological factors to PM1 at the same time, enabling their direct comparison. The model does allow for interactions between the meteorological factors, and on this basis, a separation and comparison of meteorological influences on any individual event is feasible. This is a novel aspect, as it allows to extract empirical patterns from the data set that are hard to detect using established statistical methods.*

*Setting up a forecasting tool is a possible extension of the machine-learning framework established within this study, but not the key objective here. This is why we only outline such possible applications and their usefulness at the end of the manuscript. So no, our analysis framework in its present form is not intended as a forecasting tool, and cannot be converted into one without more work. Hence, the reliability of such a forecast tool was not assessed. It is likely that the PM forecast would greatly depend on the reliability of the forecasted meteorological conditions. In its present configuration, however, our tool can determine an 'expected' level of air pollution under given meteorological conditions. By comparing this to actual observations, the effect of any source reductions (e.g. via policies) can be assessed. These points were added in L510-520.*

*The following specific changes were made in the manuscript:*
- *L2 & 3: "substantially contribute to" was changed to "substantially influence". The wording "contribute to" might indeed be misleading here, as it could sound as if meteorology actively emits pollutants.*
- *Throughout the manuscript, the wording "meteorological contribution" was changed to "meteorological influence" or removed, if not referring to the ML model (caption chapters 4.2, 4.2.1-4.2.4, L265, L294, L295, 295, 303, 317, 329, 361, 362, 376, 424, 461, 483, 486)*

- *L46: the sentence "It is therefore crucial to take atmospheric and environmental processes into account during the development of efficient pollution mitigation strategies" was removed. This point is now made clearer at the end of the introduction*
- *See changes in lines 80-85; the goal of the study is now stated more precisely and benefits in a public health context are described*
- *L476-479: ...As interactions between the meteorological variables are accounted for, the model enables the separation, quantification and comparison of their respective impacts the individual events. It is shown that ambient meteorology can substantially exacerbate air pollution. Results of this study point to a distinguished role of shallow MLHs, low temperatures and low wind speeds during peak PM1 concentrations in winter*
- *L512-515: changed to "For policy makers, the presented approach could prove beneficial in multiple ways and serve as a decision aid for air policy measures. Preventative warnings could be issued to the public if the identified meteorological conditions exacerbating air pollution are to be expected. Another application would be to attribute changes in air quality to policy measures by comparing an 'expected' level of air pollution under given meteorological conditions to actual observations (e.g., Cermak2009 and Knutti 2009), which may help…"*

**A core premise (in the abstract and elsewhere) is that we do not fully understand the contribution of meteorology to high air pollution episodes is true, however, this does justify the framework used here which omits two other key drivers (chemistry and emis-sions). Apart from a few mentions, it is not clear how are these contributions and con-sidered in this method. Are the contributions of these processes just assumed to be part of the meteorological contributions? This needs to be a lot clearer.**

*The focus of this paper explicitly lies on the analysis of the influence of meteorological conditions on PM1 concentrations. We are fully aware that meteorology alone cannot explain PM1; one of our aims is to ultimately be able to 'remove' the effect of meteorology, and retain the effects of emissions (and to a lesser degree, chemistry), which to some extent can be influenced directly by policy (this was added in L99-102). As mentioned in other answers above, pollutant concentrations have been shown to be exacerbated or decreased by certain meteorological conditions (e.g., Dupont et al., 2016). It is shown that the model is able to capture a large fraction of the occurring variation of daily PM1 concentrations, which shows that the variables chosen as inputs are indeed important drivers. Even without*

*explicitly considering emissions and chemistry, the model explains between 50-60% of the day-to-day PM1 variability. Thus, for the location and data set analysed here, the influence of meteorological variability on PM1 is at least as large as the influence of the variability of emissions and chemistry. Hence, given the key objective of this study, the presented framework is suitable for the analysis by capturing key meteorology-based processes. The detailed analysis presented in chapter 4.4 emphasizes that the temporal trends of PM1 concentrations are largely well captured.*

*Some of the meteorological parameters inherently contain information on chemistry and emissions. For example, RH, solar radiation, and temperature can influence local transformation processes, as detailed in L44-60. Temperature also contains inherent information on the strength of residential heating (L250). Wind direction indicates whether clean air from the west or more polluted air from the northeast is influencing the PM1 measurement. These mechanisms are mentioned in the introduction (L42-59) and the result section (chapter 4.2)*

*To convey these points more clearly to the reader, the following changes were made:*

- *L85: added "atmospheric", changed "determining" to "influencing"*
- *L60: Added "...while moisture in the atmosphere can stimulate secondary particle formation processes…"*
- *L136-141: added in method section (chapter 2.2): "Following the objective of this study, a set of meteorological variables is chosen as inputs for the ML model that either influence PM concentrations directly via dilution (MLH, wind speed (ws), and wet scavenging of particles (precipitation)) and particle transport (wind direction as u, v components, air pressure (AirPres)), as a proxy for emissions (e.g. from residential heating: temperature at a height of 2 m (T)), and as a proxy for transformation processes (total incoming solar radiation (TISR), relative humidity (RH), T).*

**The paper seems mostly focused on exploring the "SHapley Additive exPlanation(SHAP) values" approach and it is unclear whether a novel contribution has been made to the field of air pollution research. This paper may be better suited to a machine learn-ing journal or could be re-write to be more focused on air pollution. Either of these two options would require large changes to the current manuscript.**

*The novelty and also the advantage of the machine-learning framework is that all meteorological influences on PM1 concentrations are quantified at the same time, and interactions between the meteorological variables are captured. On this basis, their influence on any individual event can be separated and quantified (as done in*

*chapter 4.4). These aspects are novel and taken together, exceed the potential of past observation-based analyses.*

*It was not the aim of this study to explore the applicability SHAP values and it is unfortunate if this impression is conveyed by the current state of the manuscript. Therefore, extensive changes to the manuscript have been made to sharpen the scientific contribution of this manuscript and to more carefully emphasize scientific contributions.*

*Still, it is important to note here that much of the methodology chapter is dedicated to the ML algorithm and the SHAP values to make sure that the results chapter can be followed by readers not familiar with these techniques. An evaluation of the model such as in chapter 4.1 is critical to ensure that the model is able to reproduce empirical patterns.*

*Large parts of the abstract, introduction and the conclusion section were altered to shift the focus from the SHAP approach to the scientific findings. The following specific changes were made in the manuscript:*

- *L510: removed "To our knowledge, this is the first time that the SHAP-framework for explainable machine learning is applied in atmospheric sciences"*
- *Headline 3.2: added "to infer processes" to stress the purpose of the SHAP values*
- *L211: was changed to "The interactions of input features contribute to the model output and thus reflect empirical patterns that are important to deepen the process understanding."*
- *L215: deleted from the manuscript "SHAP values are a novel tool to better understand multivariate natural systems, in particular when applied in state-of-the-art machine learning models as GBRT. So far, SHAP values have been used in the fields of computer science (Antwarg et al. 2019) and medical science (Lundberg et al., 2018b; Li et al., 2019a; Lundberg et al.,2020), but have yet to be applied to study environmental systems."*
- *L96: Removed "With the use of SHAP values, a detailed insight to the decisions of the statistical model can be provided, hence allowing an advancement of previous ML approaches (Friedman, 2001; Lundberg et al., 2018a)."*
- *L508-511: Removed "The GBRT approach in combination with the SHAP regression values presented here provides an intuitive tool to assess meteorological drivers of air pollution and to advance the understanding of high pollution events by uncovering different physical mechanisms leading to high-pollution episodes."*

- *L248: added "...as suggested by Fig. 5d..." to state more clearly that this constitutes a new finding*
- *L404-405: added "The physical explanation behind this pattern would be that lacking wet deposition and low wind speeds increase particle numbers in the atmosphere, while northeastern winds advect further particles. Given that there is now a large number of particles available, the accumulation effect of a low MLH is more efficient"*
- *See also changes in L96-103, which now more clearly pinpoint the purpose of the study*

**Specific comments**

**Why has PM1 been the focus of this study, rather than the more health-relevant PM2.5 species? Also, how did the model perform at predicting PM10? Considering the omis-sion of chemistry and emissions in this study, would PM10 or PM2.5 be a better candi-date for study?**

*The available ACSM instrumentation does process only PM1 particles. PM1 is highly relevant for human health, affecting the respiratory system. Smaller particles can penetrate deeper into the lungs compared to larger particles and potentially cause more damage Studies show that health impacts of PM1 are similar (Yang et al., 2018, DOI: 10.1016/S2542-5196(17)30100-6) or worse than PM2.5 (Chen et al. 2017, DOI: 10.1016/j.envint.2018.08.027). In addition, a study by the WHO indicates that BC is a good indicator for human health, which is most prominent for particles smaller 1µm (see*
*https://www.euro.who.int/en/health-topics/environment-and-health/air-quality/publicat ions/2012/health-effects-of-black-carbon-2012)*

- *A comparison to PM10/PM2.5 is currently not feasible since no simultaneous measurements of PM1, PM2.5, PM10 and meteorological parameters at the same site are available*
- *Added to L111: "..., a highly health relevant fraction of PM including small particles that can penetrate deep into the lungs (Yang et al., 2018; Chen et al., 2017a)"*

**Line 21 - "Processes vary even within seasons" This does not read well. Of course, processes will vary within seasons.**

*Sentence was removed from the manuscript.*

**Line 24 - "likely causes an increase in local wood-burning emissions"Cause and effect seem to be muddled. Maybe the authors mean to say increases in burning emission could explain increased particulates?**

*Yes, this was the intention. To make this more clear, the sentence was changed to* *"likely triggers increased local wood-burning emissions, which increase PM1 concentrations"*

**line 25 - "The application of SHAP regression values within a machine learning frame-work presents a novel and promising way of analysing observational data sets in envi-ronmental sciences."Are there implications for what we should focus on meteorology studies or observations on? What about the implications for air-quality modelling or policy? Just presenting another tool that can be used is not a notable contribution.**

*This sentence was removed from the manuscript and replaced by* *"The identification of these meteorological conditions that increase air pollution could help policy makers to issue warnings to the public or install preemptive measures by specifically accounting for meteorological variability that influences PM1 concentrations. Furthermore, the presented framework has the potential to assess the effectiveness of air pollution measures." L8 was changed to …"Based on the model, an isolation and quantification of individual meteorological influences for process understanding is achieved..."*
*See also changes in the introduction (L98-106) and conclusion (L502-510).*

**Line 90 - How can policymakers use this information? Improve air quality models?Focus research directions? What about it is new?**

*Extensive changes in the manuscript have been made in L96-103 (see also previous answers). In addition, potential applications and the new insights were emphasized in various parts of the conclusion section (L475-480, L502-510, L515-520).*

*L482-485: changed to* *"For policy makers, the presented approach could prove beneficial in multiple ways and serve as a decision aid for air policy measures. Another application would be to attribute changes in air quality to policy measures by comparing an 'expected' level of air pollution under given meteorological conditions to actual observations (e.g.,Cermak2009 and Knutti 2009), which may help…"*

**Line 90 - Why not focus on the SIRTA region, rather than Paris, which is in completely different chemistry and emissions regime? The reader needs to be convinced that the site is representative of the Paris region.**

*The results relate to the measurement site, which is representative of the Paris region background values. This was added in L127: "**PM1 measurements are representative of background pollution levels of the region of Paris (Petit2015 et al., 2015)**"*

*Sentence was rephrased in L94* *"govern pollution concentrations at the measurement site"* *instead of "lead to high pollution events in Paris"*

**Technical comments**

**Please use sub/superscripts for chemical species throughout (e.g. SO42-, SO2,PM2.5).**

*This was adjusted accordingly.*

**Expand acronyms in sub-header titles (e.g. MLH).**

*This was adjusted accordingly.*

**Expand acronyms once per major section too.**

*Given the limited number of acronyms, the authors propose to extend them only at the first mention.*

**Meteorology-driven variability of air pollution (PM$_1$) revealed with explainable machine learning**

Roland Stirnberg[1,2], Jan Cermak[1,2], Simone Kotthaus[3], Martial Haeffelin[3], Hendrik Andersen[1,2], Julia Fuchs[1,2], Miae Kim[1,2], Jean-Eudes Petit[4], and Olivier Favez[5]

[1]Institute of Meteorology and Climate Research, Karlsruhe Institute of Technology (KIT), Karlsruhe, Germany
[2]Institute of Photogrammetry and Remote Sensing, Karlsruhe Institute of Technology (KIT), Karlsruhe, Germany
[3]Institut Pierre Simon Laplace, École Polytechnique, CNRS, Institut Polytechnique de Paris, Palaiseau, France
[4]Laboratoire des Sciences du Climat et de l'Environnement, CEA/Orme des Merisiers, Gif sur Yvette, France
[5]Institut National de l'Environnement Industriel et des Risques, Parc Technologique ALATA, Verneuil en Halatte, France

**Correspondence:** Roland Stirnberg (Roland.Stirnberg@kit.edu)

**Abstract.** Air pollution, in particular high concentrations of particulate matter smaller than 1 $\mu$m in diameter (PM$_1$), continues to be a major health problem, and meteorology is known to $^{RS:}$substantially influence atmospheric PM concentrations. However, the scientific understanding of the $^{HA:}$ways by which complex $^{HA:}$interactions of meteorological factors lead to high pollution episodes is inconclusive, as the effects of meteorological variables are not easy to separate and quantify. In this study, a novel, data-driven approach based on empirical relationships is used to characterise $^{HA:}$, and better understand the meteorology-driven component of PM$_1$ variability. A tree-based machine learning model is set up to reproduce concentrations of speciated PM$_1$ at a suburban site southwest of Paris, France, using meteorological variables as input features. $^{RS:}$Based on the model, an isolation and quantification of individual meteorological influences for process understanding is achieved $^{RS:}$ using SHapley Additive exPlanation (SHAP) regression values. $^{RS:}$ $^{RS:}$Season-specific processes influencing PM1 concentrations at the measurement site are analysed in selected high-resolution case studies.$^{RS:}$ Model results suggest that winter pollution episodes are often driven by a combination of shallow mixed layer heights (MLH), low temperatures, low wind speeds or inflow from northeastern wind directions. Contributions of MLHs to the winter pollution episodes are quantified to be on average ~5 $\mu$g/m$^3$ for MLHs below <500 m agl. Temperatures below freezing initiate formation processes and increase local emissions related to residential heating, amounting to a contribution $^{RS:}$to predicted PM$_1$ concentrations of as much as ~9 $\mu$g/m$^3$. Northeasterly winds are found to contribute ~5 $\mu$g/m$^3$ to $^{RS:}$predicted PM$_1$ concentrations (combined effects of u- and v-wind components), by advecting particles from source regions, e.g. central Europe or the Paris region. However, in calm conditions (i.e. wind speeds < ~2 m/s), the lack of dispersion leads to increased PM$_1$ concentrations by ~3 $\mu$g/m$^3$. Unusually high PM$_1$ concentrations in summer are generally lower compared to winter peak concentrations, and are characterised by a higher content of organics. Meteorological drivers of summer peak PM$_1$ concentrations are temperatures above ~25 °C (contributions of up to ~2.5 $\mu$g/m$^3$), dry spells of several days (maximum contributions of ~1.5 $\mu$g/m$^3$) and wind speeds below ~2 m/s (maximum contributions of ~3 $\mu$g/m$^3$ ). High-resolution case studies show a large variability of processes, which together lead to high PM$_1$ concentrations. $^{RS:}$

vary even within seasons. A high pollution episode in January 2016 is shown to be driven by a drop in temperature (maximum contributions of 11 $\mu$g/m$^3$), which enhances formation of secondary inorganic aerosols (SIA) and [RS:]  likely triggers increased local wood-burning emissions, which increase PM$_1$ concentrations. In contrast, during December 2016, high PM$_1$ concentrations are caused mainly by a shallow MLH and low wind speeds. It is shown that an observed decrease in pollution levels is linked to a change in wind direction, advecting cleaner, maritime air to the PM measurement site (combined contributions of u- and v-wind-components of $\sim$-4 $\mu$g/m$^3$). [RS:]  [RS:] The identification of these meteorological conditions that increase air pollution could help policy makers to issue warnings to the public or install preemptive measures by specifically accounting for meteorological variability that influences PM1 concentrations. Furthermore, the presented approach has the potential to realistically assess the effectiveness of air pollution measures.

*Copyright statement.* TEXT

**1 Introduction**

[revised manuscript text omitted]

Here, the multivariate and highly interconnected [RS:]nature of meteorology-dependent atmospheric processes influencing [RS:] local PM$_1$ concentrations [RS:]at a suburban site southwest of Paris is [RS:] analysed in a data-driven way. [RS:]Therefore, a state-of-the-art explainable machine learning model is set up to reproduce the variability of

90  PM$_1$ concentrations, thereby capturing empirical relationships between PM$_1$ concentrations and meteorological parameters. [RS:] , [RS:] The goal is to separate and quantify influences of the meteorological variables on PM$_1$ concentrations to advance [RS:]  the process understanding of the complex mechanisms that [RS:]  govern pollution concentrations at the measurement site. Localised (i.e. situation-based) and individualised attributions of feature contributions are performed using SHapley

95  Additive exPlanation regression (SHAP) values (Lundberg and Lee, 2017; Lundberg et al., 2018a, 2020), [RS:] allowing to infer meteorology-dependent processes driving PM concentrations at high temporal resolution. [RS:]  [RS:]  [RS:] Typical situations that lead to high PM$_1$ concentrations are

100  identified, serving as a decision support to policymakers to issue preventative warnings to the public if these situations are to be expected. In addition, by directly accounting for meteorological effects on PM1 concentrations, such a machine learning-based framework could help in assessing the effectiveness of measures towards better air quality.

105   [RS:] Furthermore, the proposed ML framework can be viewed as a first step towards a data-driven, prognostic tool in operational air quality forecasting, complementary to CTM approaches.

**2   Data sets**

Seven years (2012-2018) of meteorological and air quality data from the Site Instrumental de Recherche par Télédétection Atmosphérique (SIRTA, Haeffelin et al., 2005) supersite are the basis of this study. The SIRTA Atmospheric Observatory is

110  located about 25km southwest of Paris (48.713°N and 2.208°E, Fig. 1). This study focuses on day-to-day variations of total and speciated PM$_1$ [RS:], a highly health relevant fraction of PM including small particles that can penetrate deep into the lungs (Yang et al., 2018; Chen et al., 2017a). To separate diurnal effects e.g. the development of the boundary layer during morning hours (Petit et al., 2014; Dupont et al., 2016; Kotthaus and Grimmond, 2018a) from day-to-day variations of PM$_1$, mean concentrations of total and speciated PM$_1$ for the afternoon period 12-15 UTC are considered, when the boundary layer is fully

115  developed. In sections 2.1 and 2.2, the PM$_1$ and meteorological data and preprocessing steps before setting up the machine learning model are described. The applied machine learning model and data analysis techniques are presented in sections 3.1 and 3.2.

**2.1   Submicron particle measurements**

Aerosol chemical speciation monitor (ACSM, Ng et al., 2011) measurements are conducted at SIRTA in the framework of

120  the ACTRIS project. The ACSM provides continuous and near real-time measurements of the major chemical composition of non-refractory submicron aerosols, i.e., organics (Org), ammonium (NH$_4^+$), sulfate (SO$_4^{2-}$), nitrate (NO$_3^-$) and chloride (Cl$^-$).

[Figure]

**Figure 1.** Location of the SIRTA supersite southwest of Paris. © OpenStreetMap contributors 2020. Distributed under a Creative Commons BY-SA License.

A detailed description of its functionality can be found in Ng et al. (2011). Data processing and validation protocol can be found in Petit et al. (2015) and Zhang et al. (2019). In addition, black carbon (BC) has been monitored by a seven-wavelength Magee Scientific Aethalometer AE31 from 2011 to mid-2013, and a dual-spot AE33 (Drinovec et al., 2015) from mid-2013
125   onwards. Consistency of both instruments have been checked in Petit et al. (2014). Using the multispectral information, a differentiation into fossil fuel-based BC (BCff) and BC from wood burning (BCwb) is achieved (Sciare et al., 2010; Healy et al., 2012; Petit et al., 2014; Zhang et al., 2019). Here, the sum of all measured species is assumed to represent the total $PM_1$ content (see Petit et al., 2014, 2015). The consistency of ACSM and Aethalometer measurements is checked by comparing the sum of all monitored species with measurements of a nearby Tapered Element Oscillating Microbalance equipped with a Filter
130   Dynamic Measurement System (TEOM-FDMS). $^{RS:}$PM$_1$ measurements are representative of suburban background pollution levels of the region of Paris (Petit et al., 2015). As an additional input to the machine learning model, the average fraction of $NO_3^-$ of the previous day is added (NO3_frac). Pollution events dominated by $NO_3^-$ are often linked to regional-scale events, which depend on anthropogenically-influenced processes in the source regions of $NO_3^-$ precursors (Petit et al., 2017). This is approximated by the inclusion of the average fraction of $NO_3^-$ of the previous day, assuming that a high fraction of $NO_3^-$
135   indicates the occurrence of such an anthropogenically-influenced regime.

**2.2 Meteorological data**

$^{RS:}$Following the objective of this study, a set of meteorological variables is chosen as inputs for the ML model that either influence PM concentrations directly via

140   dilution (MLH, wind speed (ws), and wet scavenging of particles (precipitation)) and particle transport (wind direction as u, v components, air pressure (AirPres)), as a proxy for emissions (e.g. from residential heating: temperature at a height of 2 m (T)), and as a proxy for transformation processes (total incoming solar radiation (TISR), relative humidity (RH), T). Data are taken from the quality-controlled and 1h averaged re-analysed observation (ReObs) dataset. Further information on the instrumentation used for the acquisition of these variables is provided in Chiriaco et al. (2018). MLH is derived from automatic

145   lidar and ceilometer (ALC) measurements of a Vaisala CL31 ceilometer using the CABAM algorithm (Characterising the Atmospheric Boundary layer based on ALC Measurements, Kotthaus and Grimmond, 2018a, b). Due to an instrument failure, during the period July to mid-November 2016, SIRTA ALC measurements had to be replaced with measurements conducted at the Paris Charles de Gaulle Airport, located northeast of Paris. [RS:]A comparison of measured MLHs at SIRTA and Charles de Gaulle Airport for the available measurements in 2016 (Appendix A) shows generally good agreement, which is why only

150   minor uncertainties are expected due to the replacement.

[revised manuscript text omitted]

In agreement with results from previous studies focusing on $PM_{10}$ (Grange et al., 2018; Stirnberg et al., 2020) or $PM_{2.5}$ (Liu et al., 2018), SHAP values do not change much for MLH above ∼800–900 m, i.e. boundary layer height variations above this level do not influence submicron particle concentrations. Positive contributions of MLHs above ∼800–900 m [RS:]on predicted $PM_1$ concentrations, as visible in Fig. 6 [RS:]for some species, 
[revised manuscript text omitted]

Many of the results presented here hold true for regions other than [RS:]  the Sirta supersite and are thus beneficial for the general understanding of drivers of air pollution. This includes the [RS:] nonlinear response of $PM_1$ concentrations to changes in temperature or MLH, including their dependencies on other meteorological factors, which has potential implications for the settings of future CTMs. Furthermore, the temperature-dependency of $PM_1$ is relevant in terms of rising temperatures due to climate change [RS:] . The [RS:]  importance of wind direction highlights the role of advected pollution and emphasizes the need for large-scale measures against air pollution. [RS:]  [RS:] .

[RS:] For policy makers, the presented approach could prove beneficial in multiple ways for air quality policies. Preventative warnings could be issued to the public if the identified meteorological conditions exacerbating air pollution are to be expected. [RS:]  [RS:] Another potential future application could be the attribution of changes in air quality to policy measures [RS:] by comparing an "expected" level of air pollution under given meteorological conditions to actual observations (e.g., Cermak and Knutti, 2009),[RS:]  which may help political decision makers develop and implement effective clean-air policies. Future efforts could also combine the statistical model framework with short-term weather forecasts, which would allow to provide an air quality forecast based on the predictions of the statistical models, taking into account expected meteorological conditions. This study could be extended in the future, e.g. by including information on anthropogenic emissions or further stations down- and upwind of SIRTA, which would allow further analysis of dominant advection patterns. Furthermore, information on emissions or meteorology in the source region of air masses e.g., using satellite-based observations, might be helpful to better reproduce particle transport patterns. This could be complemented by incorporating synoptic variables, e.g., the North Atlantic Oscillation (NAO) index.

*Data availability.* SIRTA-ReOBS data can be accessed online (https://sirta.ipsl.fr/reobs.html), ACSM data are available upon request.

[Figure]

**Figure A1.** Scatterplot for MLH [m agl] measured at Sirta vs. MLH measured at Charles de Gaulle airport .

[Figure]

**Figure A2.** Histogram showing the frequency of occurrence for MLH [m agl] measured at Sirta (red) vs. MLH measured at Charles de Gaulle airport (black).

**Appendix A:  Comparison of mixed layer height (MLH) measured at Sirta and Charles de Gaulle airport**

As mentioned in section 2.2, ca. 90 missing MLH values in 2016 were replaced with measurements conducted at the Charles de Gaulle airport (see Fig. 1). Figs A2 and A1 summarize MLH values for 2016 when measurements from both sites are available (afternoon period). As shown in Fig. A1, measurements at both sites generally agree well, except for some outliers. Spearman's rank coefficient is significant (p-value $< 0.05$) and has a value of 0.51.

A comparison of the frequency of occurrence is shown as histogram in Fig. A1 and indicates good agreement as well.

[revised manuscript text omitted]

---

## Author Response (AR2)

**2nd Rebuttal ACPD**

**BOLD** = reviewer comment
*Italic* = answer to reviewer comment
Red = highlighted changes in manuscript

**The paper presents a machine learning approach to assess the impact of several meteorological features on air quality in Paris metropolitan area. A tree-based machine learning algorithm is used for modelling and a Shapley Additive Explanation is applied to interpret the resulting models. This is a quite interesting study that requires, however, major revisions before a possible publication.**
**In the next version, the authors must address all the points as follows:**

**1) The abstract should be improved. On one hand, the it is too long. On the other hand, important information is missing, such as the accuracy/performance of the models.**
*Thank you for your evaluation. We carefully considered your concerns and took great care to adequately address these.*
*In the abstract, the following changes were made:*
*L4: removed "as the effects of meteorological variables are not easy to separate and quantify."*
*L7-11: added "The model is able to capture the majority of occurring variance of mean afternoon total PM$_1$ concentrations (coefficient of determination ($R^2$) of 0.58), with model performance depending on the individual PM$_1$ species predicted"*
*L11/12: shortened the sentence to make it more concise*
*L24/25: shortened the sentence*
*L18/19: deleted sentence*
*L24/28: replaced sentences with "High-resolution case studies are conducted showing a large variability of processes that can lead to high pollution episodes."*
*L33/34: changed to "...to adapt policy measures, issue warnings to the public, or to assess the effectiveness of air pollution measures."*

**2) Even if the approach is interesting, it is a quite local study (Paris area). I would like that the authors provide a further discussion about the general impact of their work. In other words, you should discuss to which extend the study has implications in other urban areas worldwide, inclusively cities with more complex terrains than Paris.**
We agree that further discussion about the implications of this local study for other urban and suburban regions increases the value of our findings. We have added the following thorough discussion on this (L490-509) and feel that this has helped improve the manuscript. "*The presented findings refer to the SIRTA supersite but the results are nevertheless transferable to other regions as well. For example, the importance of temperature-induced particle formation processes have been shown for the U.S.A. (Dawson et al., 2007), Europe (Megaritis et al.,*

*2014), and China (Wang et al., 2016). Hence, it is likely that the detailed, species-dependent disclosure of the nonlinear relationship between temperature and PM1 of this study holds for other urban and suburban areas. This has implications for the PM concentrations in the context of climate change. The empirical perspective of the current study complement to the findings of various modelling studies (Dawson et al., 2007; Megaritis et al., 2013, 2014; Sá et al., 2016; Doherty et al., 2017).*

*Furthermore, the impact of shallow MLHs on PM1 concentrations investigated here is comparable to results found in a previous, regional-scale study over central Europe that highlighted the dominant role of MLH on PM10 concentrations (Stirnberg et al. 2020).*

*The importance of wind direction highlights the role of advected pollution by remote, highly polluted urban or industrial hotspots. In general, the interpretation of pollution advection patterns requires knowledge on source regions and terrain. Here, the Paris agglomeration is a major source of pollutants while the relatively flat terrain allows unimpeded advection of air masses.*

*Urban areas in a more complex terrain would likely be affected by slightly different and possibly more complex mechanisms., such as terrain- and meteorology-dependent air stagnation events (Wang et al. 2018) as well as orography driven wind and precipitation patterns (Rosenfeld et al., 2007).*

*Still, given the task of disentangling the impact of the various meteorological drivers on air quality is already a complex scientific subject, a continental, flat terrain city such as Paris was chosen as the subject area precisely to exclude other factors (such as orographic flow, or sea breeze) that would add further complexity. Certainly, the methods developed here could be transferred to more urban areas in more complex settings in the framework of future studies.*

**3) Lines 163-164. The sentence "Note that PM1 data is not normally distributed, i.e. there is more data available for mid-range concentrations" is awkward. Is it not a characteristic of a normal distribution to have more data in mid-range? Please, clarify this sentence.**

*Yes, this sentence was indeed awkwardly formulated. It was changed to "Note that PM1 data is not uniformly distributed, i.e. there is more data available for mid-range PM1 concentrations."*

**4) Section 4.1. The ten models you are talking about are not clear. More details must be provided regarding what is modelled by each model and the acronyms BCwb, BCff, etc… have to be defined.**

*The ten models are mentioned in L177. To make this clearer in the manuscript, we added in L208: "The performance of the species and total PM1 models, each with ten model iterations (of which each has different hyperparameters),..."*

*Added in caption of Fig. 3:*

- *(Org: organics, NH4: ammonium, SO4: sulfate, NO3: nitrate, Cl: chloride, BCff: black carbon from fossil fuel combustion and BCwb: black carbon from wood burning)"*
- *...between the ten model iterations."*

**5) Section 4.2. Why did you focus on temperature, MLH and wind direction, only? Considered that NO3 fraction and Wind Speed are also strong drivers, why did you skip a deep interpretation of the effect of these variables, as well?**

- *As pointed out in chapter 4.1, the NO3 fraction is found to vary with PM1 total mass concentrations. Lower wind speed generally leads to higher particle concentrations. Temperature, MLH and wind direction, on the other hand, require an in-depth analysis of the different PM1 species, as changes of these variables causes nonlinear responses in PM1 predictions, which vary also between species.*
- *To make this clearer in the manuscript, L221 was changed: "Lower wind speeds generally lead to higher particle concentrations (see Fig. B2) due to a lack of dispersion (Sujatha2016). Temperature, MLH and wind direction require an in-depth analysis, as changes of these variables cause nonlinear responses in PM1 predictions, which vary also between species. "*
- *Added "(see Fig. B)" in L 221*
- *Plots showing the influence of wind speed and NO3 fraction, respectively, were added in the appendix*

**6) Line 247. Change "Fig. 6" to "Fig. 5-7".**
*Done*

**7) Lines 282-285. You noticed that north/north-eastern winds increase air pollution and you conclude that this pollution should come from Paris, which is located north-eastern from SIRTA. Did you confirm this assumption by analysing wind data from the Airport Charles de Gaulle? If the hypothesis is true, bad air should come from south/south-western in this case. Right?**
*No PM1 data is available to us from Charles de Gaulle (we used only MLH data for a limited period). This is why we point out that advected particles come from the Paris region and/or continental Europe, as suggested by previous studies. For example, results by Petit et al. 2017, who compared PM1 concentrations at different locations in France during a high-pollution episode suggest that long-range transport of polluted air from continental Europe can be a dominating driver. Our approach is not well suited to distinguish between long-range advection from continental Europe or the Paris region as we focus only on one station.*

**8) Section 4.2.4. It is not clear which species you are interested in for the interaction analysis. Is it PM1, only? Please, be more specific.**
*L310: added "Pairwise interaction effects, where the effect of a specific predictor on the total PM$_1$ prediction is dependent on the state of a second predictor, are analysed in the model."*

**9) Figure 8. How do you explain the red cluster on the top-right corner of the right panel? In other words, how do you explain that high wind speed and high MLH tend to increase the Shap values?**
*A physical explanation could be the more effective transport of SO4 and its precursor SO2 and ammonium nitrate under high-MLH-conditions and stronger winds (Pay et al., 2012).*
*L325: added "High MLHs in combination with high wind speeds, however, increase SHAP values. A physical explanation of this pattern could be the more effective transport of SO4 and*

*its precursor SO2 as well as ammonium nitrate under high-MLH conditions and stronger winds, and increased formation rates of secondarily formed particles as mentioned in chapter 4.2.2"*

**10) Section 4.4, 1st paragraph. This paragraph should be reorganized. You give several details about Figs 11-16, which are irrelevant here (lines 361-363). On the other hand, this information misses in the caption of these respective figures.**
*Paragraph was moved to the caption of Fig. 11.*

**11) Line 393-395. You explain the high pollution in terms of weak "north-north-easterly winds, i.e. a regime of low ventilation". However, it can also be a weak wind that brings pollution from Paris. Please, comment on this point.**
*Yes, it is true that advection also plays a role here (can be deduced from u-wind SHAP values). Since wind speeds are low and MLHs are also low, thus impeding effective transport of air masses, advection is expected to play a minor role.*
*L401: added … "and possibly some advection of polluted air from the Paris region"*

**12) Figures 11-14. The quality of these figures must be improved. First, the legend is too small. Second, indexes a)-h) are missing in Fig. 11. Third, it is not straight-forward to understand the matching between the bar/scatter plots and the right/left side of the Y-axes. Finally, you do not describe in the caption how the predicted vs the observed PM1 are represented. So, the caption needs to be improved, based on my comment 10, as well.**
- *Legend sizes in Figs 11-14 were increased*
- *Indexes a)-h) were added to Fig. 11*
- *The caption was extended*
- *An explanation for left and right y-axes was added*

**13) Conclusion. We understand that your models do a better job in Winter and Summer than in Spring. So, what about Fall? Why do you not present data for this period? Is it also more difficult to do a good prediction at this season? If, yes, can we conclude that the approach is less suitable for the midseason, maybe because the meteorological conditions are less "extreme" (e.g., average temperature)?**
- Fall is very similar to winter in terms of prediction accuracy and drivers of high-pollution situations, except for the occurrence of temperatures below zero, which have a distinct positive influence on PM1 concentrations in winter.

[Figure]

[Figure]

- The reduced prediction accuracy in spring shown in FIg. 14 is due to the exceptional character of this pollution episode. In general, we decided to focus on the most extreme seasons (summer vs. winter) and contrast these in chapter 4.3 and 4.4. T

**14) Lines 474-475. Which evidences support this quite strong statement. More arguments are expected, especially to address my comment 2).**
*Please see changes referring to your 2nd comment.*

**15) Conclusion, last paragraph. This paragraph is very redundant. We understood at the first sentence that a meteorological prediction is important if we want to use your approach. However, it seems that you repeat the same idea again and again. The proof is the fact that the word "expected" appears three times in the next sentences. This last paragraph must be improved by reorganizing its structure.**

*The last paragraph was restructured. The idea of this paragraph was actually to convey three different ideas, i.e., three different possible approaches of the presented model approach.*

- *Preventative warnings to the public based on knowledge of meteorological conditions exacerbating air pollution*
- *Quantitative analysis of the effects of air pollution measures*
- *Pollution forecast based on short-term weather forecasts*

*These different ideas are hopefully conveyed more concisely now.*